# Investigation of the anti-tumor mechanism of tirabrutinib, a highly selective Bruton's tyrosine kinase inhibitor, by phosphoproteomics and transcriptomics

Ryohei Kozaki◉*◔, Tomoko Yasuhiro◔, Hikaru Kato◉, Jun Murai◉, Shingo Hotta, Yuko Ariza, Shunsuke Sakai, Ryu Fujikawa, Takao Yoshida

Discovery and Research, Ono Pharmaceutical Co., Ltd, Osaka, Japan

◔ These authors contributed equally to this work.
* kozaki@ono-pharma.com

**Data Availability Statement:** There are no ethical nor legal restrictions on sharing data. We have included the minimal anonymized dataset in

## Abstract

Tirabrutinib is a highly selective Bruton's tyrosine kinase (BTK) inhibitor used to treat hematological malignancies. We analyzed the anti-tumor mechanism of tirabrutinib using phosphoproteomic and transcriptomic methods. It is important to check the drug's selectivity against off-target proteins to understand the anti-tumor mechanism based on the on-target drug effect. Tirabrutinib's selectivity was evaluated by biochemical kinase profiling assays, peripheral blood mononuclear cell stimulation assays, and the BioMAP system. Next, *in vitro* and *in vivo* analyses of the anti-tumor mechanisms were conducted in activated B-cell-like diffuse large B-cell lymphoma (ABC-DLBCL) cells followed by phosphoproteomic and transcriptomic analyses. *In vitro* kinase assays showed that, compared with ibrutinib, tirabrutinib and other second-generation BTK inhibitors demonstrated a highly selective kinase profile. Data from *in vitro* cellular systems showed that tirabrutinib selectively affected B-cells. Tirabrutinib inhibited the cell growth of both TMD8 and U-2932 cells in correlation with the inhibition of BTK autophosphorylation. Phosphoproteomic analysis revealed the down-regulation of ERK and AKT pathways in TMD8. In the TMD8 subcutaneous xenograft model, tirabrutinib showed a dose-dependent anti-tumor effect. Transcriptomic analysis indicated that IRF4 gene expression signatures had decreased in the tirabrutinib groups. In conclusion, tirabrutinib exerted an anti-tumor effect by regulating multiple BTK downstream signaling proteins, such as NF-κB, AKT, and ERK, in ABC-DLBCL.

## Introduction

The B-cell receptor (BCR) signaling pathway plays a key role in the development of hematologic B-cell-derived malignancies, including chronic lymphocytic leukemia (CLL) [1], diffuse large B-cell lymphoma (DLBCL) [2], mantle cell lymphoma (MCL) [3], and other non-Hodgkin's lymphomas [4]. Bruton's tyrosine kinase (BTK), a cytoplasmic TEC family kinase with multifaceted functions within the BCR pathway [5], is a key therapeutic target in the treatment of B-cell malignancies [6,7].

Supporting Information as listed below: S2 Table. Complete KINOMEscan dataset. S3 Table. Processed version of the MaxQuant phosphosite table of the experiments with tirabrutinib-treated TMD8 cells. S4 Table. Processed version of the MaxQuant phosphosite table of the experiments with tirabrutinib-treated U-2932 cells. In addition, the microarray data are available from the Gene Expression Omnibus (GEO; GSE 210284). The other datasets used and/or analyzed during the current study are available from the corresponding author on reasonable request.

**Funding:** Ono Pharmaceutical Co., Ltd. (https://www.ono-pharma.com/) funded the study and medical writing support. The funder provided support in the form of salaries for all authors and was involved in the decision to publish the manuscript, but was not involved in study design, data collection and analysis, or preparation of the manuscript.

**Competing interests:** This study was funded by Ono Pharmaceutical Co., Ltd. (Osaka, Japan). All authors are employees of Ono Pharmaceutical Co., Ltd. Some of the authors are inventors of patents filed by Ono Pharmaceutical Co., Ltd. in relation to this work: WO2015/067586 ("Combination therapy of an afucosylated CD20 antibody with a BTK inhibitor", Tomoko Yasuhiro); WO2015/146159 ("Prophylactic agent and/or therapeutic agent for diffuse large B-cell lymphoma", Ryohei Kozaki); WO2016/209961 ("Combination therapies for treating B cell malignancies", Tomoko Yasuhiro and Ryohei Kozaki); WO2017/059224 ("Combination of a BTK inhibitor and a checkpoint inhibitor for treating cancers", Tomoko Yasuhiro); and WO2019/208805 ("Preventive and/or therapeutic agent for autoimmune disease comprising compound having BTK inhibitory activity as active ingredient", Yuko Ariza).

The first-generation BTK inhibitor ibrutinib was first approved in the US for MCL treatment in 2013 [8] and is now also indicated for use in patients with CLL, small lymphocytic lymphoma, Waldenström macroglobulinemia, and marginal zone lymphoma [9]. However, limitations include off-target toxicity and the development of resistance [10–12].

Second-generation BTK inhibitors (acalabrutinib and zanubrutinib) are more kinase selective [13,14]; because of their selectivity, they are expected to be superior to ibrutinib in terms of safety [15]. Clinical studies have shown that patients with CLL who are intolerant to ibrutinib can benefit from treatment with acalabrutinib [16], and a recent phase 3 head-to-head trial demonstrated that acalabrutinib had non-inferior efficacy and fewer adverse events (AEs) compared with ibrutinib [17]. Similar positive results have also been reported for zanubrutinib [18].

Tirabrutinib is another second-generation BTK inhibitor with a potent, highly selective, irreversible profile. It is approved in Japan for the treatment of relapsed/refractory primary central nervous system lymphoma, Waldenström macroglobulinemia, and lymphoplasmacytic lymphomas [19].

Some non-clinical studies have shown the anti-tumor effect and anti-tumor mechanism of BTK inhibitors in ABC-DLBCL cell lines [20,21]. However, a comprehensive analysis by both phosphoproteomics and transcriptomics in ABC-DLBCL treated with BTK inhibitors exploring anti-tumor mechanisms has not been reported.

To elucidate the anti-tumor mechanisms of molecular-targeted drugs based on the regulation of a specific target molecule, it is critical to use appropriate and selective tools and compounds. An unexpected anti-tumor mechanism related to the off-target molecules regulated by molecular-targeted drugs is often reported [22,23]. Therefore, we carefully checked the potential off-target kinases for tirabrutinib using both biochemical and cellular methods before exploring its anti-tumor mechanisms in ABC-DLBCL cell lines. Thus, the objective of this *in vitro* and *in vivo* study was to analyze the anti-tumor mechanism of tirabrutinib, a highly selective BTK inhibitor in ABC-DLBCL, using phosphoproteomic and transcriptomic techniques.

## Materials and methods

### Reagents and cell cultures

Tirabrutinib, ibrutinib, and acalabrutinib were obtained from the Department of Drug Discovery Chemistry, Ono Pharmaceutical Co., Ltd. (Osaka, Japan). Zanubrutinib and refametinib were purchased from MedChemExpress (Princeton, NJ, USA). BMS-345541 and MK-2206 were purchased from Selleck Chemicals (Houston, TX, USA) and Shanghai Haoyuan Chemexpress (Shanghai, China), respectively. Human peripheral blood mononuclear cells (PBMC) were obtained from VERITAS Corporation (Tokyo, Japan; Lot No. A1929). The activated B-cell-like (ABC)-DLBCL cell lines TMD8 and U-2932 have been described previously [24,25]; TMD8 cells were obtained from the Tokyo Medical and Dental University (Tokyo, Japan), and U-2932 cells from Deutsche Sammlung von Mikroorganismen und Zellkulturen GmbH (DSMZ, Braunschweig, Germany). PBMC and these cell lines were maintained in a culture medium, namely RPMI medium 1640 (Invitrogen Corporation) containing 10% v/v inactivated fetal bovine serum (Invitrogen Corporation) and 1% v/v penicillin-streptomycin liquid (Invitrogen Corporation) at 37°C, 5% $CO_2$/95% air.

### KINOMEscan analysis

Kinase profiling was carried out by DiscoverX (Eurofins DiscoverX Products, LLC., Fremont, CA, USA), using the KINOMEscan scanMAX platform as previously described [26]. Briefly, this is a biochemical kinase profiling assay that measures drug binding using a panel of DNA-

tagged kinases. In the current analysis, four test compounds were evaluated at 300 nM. Compounds binding to the kinase active site reduce the amount of kinase captured on a solid support of an immobilized ligand. The captured kinases are quantified using quantitative polymerase chain reaction (qPCR) that detects the associated DNA label. The strength of compound binding is determined based on its ability to block kinase-ligand binding, and the results were reported as percent control (%CTRL), which was the percentage of each kinase that remained bound to the immobilized ligand.

### *In vitro* kinase assays

Half-maximal inhibitory concentrations ($IC_{50}$) and selectivity of each test compound were calculated for a selection of 14 kinases using an optimized mobility shift assay (MSA) for FYN, LYNa, LCK, ERBB4, and CSK; the Invitrogen Z'-Lyte kinase inhibition assay for BTK, BLK, BMX, EGFR, ERBB2, ITK, JAK3, and TXK; and the Invitrogen LanthaScreen Eu kinase binding assay for TEC. A single or three independent tests (from which the average value was calculated) were conducted, with the selectivity based on a reference value of 1 for the $IC_{50}$ of BTK.

The MSA was carried out by Carna Biosciences Inc. (Kobe, Japan), and the details are described elsewhere [27]. In brief, compounds were dissolved in DMSO to desired concentrations. The prepared DMSO solutions were diluted in assay buffer to yield a final concentration of 1% DMSO for each compound. The kinase assays were carried out using ATP at its Km for each kinase, i.e., 50, 10, 10, 25, and 5 μM for FYN, LYNa, LCK, ERBB4, and CSK, respectively. Each test compound was preincubated with kinase solution for 30 or 60 min (30 min for FYN, LYNa, LCK, and 60 min for ERBB4 and CSK) at room temperature before adding the substrate/ATP/metal solution and incubating the kinase reaction for 1–5 h (1 h for FYN, LYNa, LCK, ERBB4, and 4 or 5 h for CSK). The inhibition of kinase activity by each compound was calculated as follows: inhibition (%) = [1 –(A–B) / (C–B)] × 100, where A is the response with compound, B is the background response with no kinase, and C is the response with vehicle (1% DMSO). The $IC_{50}$ value of each compound was calculated by interpolation on a log-concentration-response curve fitted with a four-parameter logistic equation.

The Invitrogen Z'-Lyte kinase inhibition assay and the Invitrogen LanthaScreen Eu kinase binding assay were carried out by Thermo Fisher Scientific (Waltham, MA, USA), following the manufacturer's instructions [28,29]. In all the assays, each test compound was preincubated with peptide/kinase mixture or kinase/antibody mixture for 60 min at room temperature before adding the ATP solution or AlexaFluor labeled Tracer to initiate the 60-min incubation of the reactants.

### PBMC stimulation assays

Human PBMCs were treated with vehicle (0.1% DMSO) or various concentrations of tirabrutinib for 10 min at 37°C and 5% $CO_2$/95% air, then stimulated with either anti-Human IgM-UNLB (Southern Biotech, #2022–01, referred to as anti-IgM hereafter; final concentration: 1 μg/mL) or Dynabeads Human T-Activator CD3/CD28 (Invitrogen Corporation, #11131D, referred to as anti-CD3/CD28 hereafter; final concentration: $1.8 \times 10^6$ beads/mL) for 22 or 16 h to evaluate the effect on B- or T-cell activation, respectively. After culturing, the stimulated cells were stained with fluorescently labeled CD3, CD20, and CD69 antibodies. B-cells (identified by gating CD20-positive and CD3-negative cells) and T-cells (identified by gating CD3-positive and CD20-negative cells) were analyzed by flow cytometry to monitor the expression of CD69. The CD69 inhibition rate (%) in the tirabrutinib-treated group was calculated versus the vehicle group, and the concentration at which tirabrutinib showed 50% CD69

inhibition (IC$_{50}$, nM) and its 95% confidence interval (CI) were estimated by nonlinear regression analysis, using a four-parameter logistic model. GraphPad Prism version 5.04 (GraphPad Software, Inc., San Diego, CA, USA) and its base settings were used for initial values and convergence conditions.

## BioMAP profiling

The biological selectivity of tirabrutinib at concentrations of 120, 370, 1100, and 3300 nM was evaluated in 12 *in vitro* BioMAP Diversity PLUS systems (Eurofins DiscoverX Products, LLC) according to the service provider's protocol, as previously described [30]. In brief, cultures with one or more primary cell types from healthy human donors were treated with sets of stimulators specific to the cell subsets, and the optimized biomarker readouts for each system were measured by ELISA and other techniques, as listed in **S1 Table**. The assays included positive and negative controls as well as vehicle control. Results are presented as the log-transformed ratio of biomarker readouts for tirabrutinib-treated samples to vehicle controls.

## Cell viability assay

TMD8 and U-2932 cells were treated with vehicle (0.1% DMSO) or various concentrations of tirabrutinib in untreated 96-well plates ($1 \times 10^4$ or $2 \times 10^4$ cells/well for TMD8 or U-2932, respectively). After incubation for 72 h at 37˚C and 5% CO$_2$/95% air, relative light units (RLU) were measured with the CellTiter-Glo Luminescent Cell Viability Assay (Promega, Madison, WI, USA). The cell growth inhibition rate (%) in the tirabrutinib-treated group was calculated versus the rate in the vehicle group, and the concentration at which tirabrutinib showed 50% growth inhibition (IC$_{50}$, nM) was estimated by nonlinear regression analysis using a two-parameter logistic model or a Sigmoid Emax model for TMD8 and U-2932, respectively. GraphPad Prism (version 5.01J or 5.04; GraphPad Software, Inc., San Diego, CA, USA) and its base settings were used for initial values and convergence conditions.

## Western blot analyses

TMD8 and U-2932 cells were treated with vehicle (0.1% DMSO) or different concentrations of tirabrutinib and incubated for 1 or 4 h at 37˚C, 5% CO$_2$/95% air. After culturing, cells were harvested and lysed with cell lysis buffer (Cell Signaling Technology, Inc.) containing 1 mM AEBSF (Calbiochem, #101500). As a positive control of BTK autophosphorylation for western blot analysis, TMD8 cells were treated with hydrogen peroxide (H$_2$O$_2$; final concentration: 6 mM) for 10 min, harvested as a control sample, and then lysed with the cell lysis buffer. Tyrosine 223 (Tyr-223), the site of BTK autophosphorylation, has been shown to be phosphorylated by H$_2$O$_2$ stimulation [31].

Each mixture of cells and cell lysis buffer was incubated on ice for 5 min, then centrifuged at $13400 \times g$ and 4˚C for 10 min. Total lysates were loaded on 4%–12% SDS-PAGE (Bio-Rad) and transferred to a polyvinylidene difluoride (PVDF) membrane (Immobilon-P Transfer Membrane, Millipore Corporation). The blot was incubated in Tris-buffered saline with Tween 20 (TBS-T, Cell Signaling Technology, Inc.) containing 2 w/v% ECL Advance Blocking Agent (GE Healthcare Life Sciences, b) for 1 h at room temperature. The blocked PVDF membranes were immersed in TBS-T and washed three times with shaking at room temperature for 10 min. The membrane was incubated with primary antibody in TBS-T containing 2 w/v% ECL Advance Blocking Agent with gentle agitation overnight at 5˚C in a cold storage room or for 1 h at room temperature. Rabbit monoclonal anti- phosphorylated BTK (Tyr-223) antibody (p-BTK; Novus Biologicals, #NB100-79907), rabbit monoclonal BTK antibody (Cell Signaling Technology, Inc., #3533), rabbit polyclonal phosphorylated AKT (Ser-473) antibody (p-

AKT; Cell Signaling Technology, Inc., #9271), rabbit monoclonal AKT (pan) (C67E7) antibody (Cell Signaling Technology, Inc., #4691), rabbit polyclonal phosphorylated p44/42 MAPK (ERK1/2) (Thr-202/Tyr-204) antibody (p-ERK1/2; Cell Signaling Technology, Inc., #9101), rabbit monoclonal p44/42 MAPK (ERK1/2) (137F5) antibody (Cell Signaling Technology, Inc., #4695), rabbit monoclonal phosphorylated PLCγ2 (Tyr-759) (E9E9Y) antibody (p-PLCγ2; Cell Signaling Technology, Inc., #50535), rabbit monoclonal PLCγ2 (E5U4T) antibody (Cell Signaling Technology, Inc., #55512), mouse monoclonal phosphorylated IκBα (Ser-32/36) (5A5) antibody (p-IκBα; Cell Signaling Technology, Inc., #9246), rabbit monoclonal IκBα (44D4) antibody (Cell Signaling Technology, Inc., #4812), rabbit polyclonal phosphorylated PKCβ (phospho-Thr-641) antibody (p-PKCβ; Signalway Antibody LLC, #11172), rabbit monoclonal IRF4 (D9P5H) antibody (Cell Signaling Technology, Inc., #15106), rabbit monoclonal BCL6 (D65C10) antibody (Cell Signaling Technology, Inc., #5650), rabbit monoclonal c-MYC (D84C12) antibody (Cell Signaling Technology, Inc., #5605), and rabbit monoclonal GAPDH (14C10) antibody (Cell Signaling Technology, Inc., #2118) were used. The PVDF membranes were immersed in TBS-T and washed three times with shaking at room temperature for 10 min, then incubated with secondary antibody with shaking for 1 h at room temperature. The secondary antibody used was anti-rabbit-IgG horseradish peroxidase (HRP)-linked antibody (Cell Signaling Technology, #7074) or goat anti-rabbit immunoglobulins/HRP (Dako, P0448). The PVDF membranes were washed three times with shaking at room temperature for 10 min and were treated with ECL$^{TM}$ Advance Western Blotting Detection Kit (GE Healthcare Life Sciences, #RPN2135) for chemiluminescent detection according to the manufacturer's instruction. Lumino-image analyzer (LAS-1000 Plus system, Fuji Photo Film Co., Ltd.) and Image Reader LAS-1000 Lite version 1.3 (Fuji Photo Film Co., Ltd.) were used to visualize bands. The densitometry analysis described the ratio of phosphorylated to total BTK protein. The IC$_{50}$ value (nM), at which 50% of BTK autophosphorylation is inhibited compared with vehicle, was calculated by nonlinear regression analysis using a four-parameter logistic model using GraphPad Prism Ver. 5.04 (GraphPad Software, Inc.).

## Phosphoproteomic analyses

The analysis was carried out by Evotec AG (Munich, Germany) as previously described [32]. In brief, two replicate SILAC experiments were carried out for TMD8 and U-2932 with inverted labeling schemes to ensure reproducibility of phosphoproteome quantification. For differential SILAC encoding, both cells were grown in arginine- and lysine-deficient culture medium containing light (Arg0/Lys0) or heavy (Arg10/Lys8) isotopic forms of L-arginine and L-lysine (SILAC medium). After culturing for 10 days to ensure complete proteome labeling, cells were seeded in fresh SILAC medium and incubated for 24 h. The cells were treated with either vehicle (0.1% DMSO) or tirabrutinib at 1 µM for 1 h. After treatment, cells were lysed and equal protein amounts from differentially SILAC-encoded and -treated cells were digested followed by peptide fractionation by strong cation exchange chromatography, followed by immobilized metal affinity chromatography phosphopeptide enrichment. For mass spectrometric analysis, phosphopeptide-enriched samples were loaded onto a reversed phase analytical column, resolved by an acetonitrile gradient using a nanoflow HPLC system and directly electrosprayed via a nanoelectrospray ion source into an LTQ-Orbitrap Velos mass spectrometer (Thermo Fisher Scientific). For data processing, all raw files acquired in this study were collectively processed with the MaxQuant software suite (version 1.2.0.18) for peptide and protein identification and quantification using a human UniProt database (version 12 2010). A false discovery rate (FDR) of 0.01 was selected for protein and peptide identification, and a posterior error probability of ≤0.1 for each peptide-to-spectral match was required. The

resulting list of phosphosites exported from the MaxQuant software was filtered for class-I sites (only serine, threonine, or tyrosine assignments with a localization probability of ≥0.75) [33] and used for all subsequent biostatistical analysis. To identify significantly regulated phosphosites on the basis of biological reproducibility, the individual phosphosite ratios were divided by each other. Gaussian regression analysis on the histogram plot of these log2-transformed ratios was performed using Sigmaplot (version 10.0; Systat Software Inc.). The obtained values for the mean and standard deviation across the whole quantitative dataset were used to determine thresholds for significant regulation of at least ±2.5 σ. The applied ratio thresholds for phosphoregulation were as follows: for TMD8 and U-2932, >1.472 and >1.506, respectively, for upregulated sites, and <0.679 and <0.664, respectively, for downregulated sites. Furthermore, only phosphosites conforming to this criterion and showing consistent regulation in both biological replicates were considered as regulated.

## RNA isolation and RT-qPCR

TMD8 cells were treated with vehicle (0.1% DMSO) or tirabrutinib at 1 μM and incubated for 24 h at 37°C, 5% $CO_2$/95% air. After culturing, total RNA was isolated from cells using the RNeasy Plus Mini Kit (Qiagen), then cDNA was prepared from total RNA using the Super-Script VILO cDNA Synthesis Kit (Invitrogen) following the kit instructions. For quantification of each mRNA expression level, PCR was performed using TaqMan Fast Universal PCR Master Mix (Thermo Fisher Scientific) under the following conditions with the AriaMx Real Time PCR System (Agilent Technologies): 95°C for 3 min, followed by 40 cycles at 95°C for 5 s and 60°C for 20 s with continuous fluorescence measurement. All PCRs were performed in triplicate, and the mRNA expression of GAPDH was used as an internal control. The results (fold changes of the threshold cycle [Ct] relative to the control group) were obtained using the $2^{-\Delta\Delta Ct}$ method. The TaqMan probes for *IRF4* (Assay ID: Hs00180031_m1), *BCL6* (Assay ID: Hs00153368_m1), *MYC* (Assay ID: Hs00153408_m1), and *GAPDH* (Assay ID: Hs02786624_g1) were purchased from Thermo Fisher Scientific. Statistical tests were performed with the SAS 9.4 (SAS Institute Japan Ltd., Tokyo, Japan) TS1M4-based EXSUS system version 10.1.3 (CAC Corporation, Tokyo, Japan). The t-test or Dunnett test was used to compare mRNA expression in the DMSO- and compound-treated groups. The statistical tests were two-sided with a 5% significance level.

## Animals and TMD8 xenograft model

The present study was conducted in compliance with the Guidance for Animal Experiments, the Ethical Standards for Experiments using Human Tissues, and the Standards for Safety Management of Pathogens established by Ono Pharmaceutical Co., Ltd. Female C.B-17/Icr-scid/scidJcl mice (hereafter referred to as severe combined immunodeficient [SCID] mice) were obtained from CLEA Japan, Inc. (Tokyo, Japan). The age at the beginning of the experiment was 6 weeks. To establish the TMD8 xenograft model, TMD8 cell suspension (0.1 mL, $1 \times 10^8$ cells/mL) was subcutaneously injected into the right flank of SCID mice under pentobarbital anesthesia. Tumor volume was calculated from the day after implantation and mice were assigned to groups when most of them showed tumor volumes of 50–250 mm³. Tirabrutinib at 1, 3, or 10 mg/kg, suspended in a 0.5 w/v% methyl cellulose 400 cP solution (Wako Pure Chemical Industries, Ltd., Osaka, Japan) or vehicle (0.5% methyl cellulose), was administered orally twice daily for 21 days, starting on day 0 (the group assignment day). Tumor diameters and body weight were measured every 3 or 4 days after group assignment. The tumor growth inhibition rate (%) in the tirabrutinib-treated groups, versus the vehicle group, was calculated based on tumor volume. Statistical tests were performed with the SAS 9.2 (SAS

Institute Japan Ltd., Tokyo, Japan) TS2M3-based EXSUS system version 7.7.1 (CAC Corporation, Tokyo, Japan). The Dunnett test was used to compare tumor volume in the tirabrutinib- and vehicle-treated groups, and linear regression analysis was used to study dose response in the tirabrutinib groups. The statistical tests were two-sided with a 5% significance level. A significant difference from vehicle in at least one tirabrutinib group and significant slope in the linear regression line were considered to indicate a dose-dependent effect.

### *In vivo* pharmacodynamics analysis

The TMD8 xenograft model mice after 21 days twice daily (BID) treatment with vehicle or various doses of tirabrutinib were used for the analysis. One hour after the final dosing, mice underwent laparotomy under isoflurane anesthesia and were sacrificed by exsanguination; tumors were collected. After measuring the wet weight, each tumor was placed on a 70-μm cell strainer (BD Falcon, Franklin Lakes, NJ, USA) placed in one well of a six-well plate on ice. After adding an amount of precooled culture medium, tumor cells were detached. The cell suspension was transferred to a tube and centrifuged for 8 min at $400 \times g$. Supernatant was excluded by aspiration, and cells were resuspended in culture medium. The tumor suspension was diluted with culture medium to a cell density of $1.0 \times 10^6$ cells/mL and pipetted into a 96-well plate. Cells were treated with phosphatase inhibitor cocktail 2 (PPTI; Sigma-Aldrich) for 30 min at 37˚C, 5% $CO_2$/95% air, and then either treated with PBS (unstimulated control) or $H_2O_2$ solution (final concentration: 3 mM) was added. Cells were incubated for a further 10 min. Fix Buffer 1 (BD Biosciences, Franklin Lakes, NJ, USA) was added to each well, and the plate was centrifuged for 8 min at $400 \times g$ at room temperature; the supernatant was removed by decantation. Each sample was washed twice with Perm Wash Buffer (BD Biosciences) before the addition of rabbit monoclonal p-BTK antibody (Tyr-223) (Novus Biologicals, #NB100-79907) diluted 500-fold with Perm Wash Buffer and 30-min incubation in the dark at room temperature. Cells underwent two rounds of Perm Wash buffer washes and centrifugation before the addition of the secondary antibody (100 μL Alexa Fluor 488 [Fab'] 2 fragments of goat anti-rabbit IgG [H+L]; Thermo Fisher Scientific) diluted 5,000-fold with Perm Wash buffer. After a 30-min incubation in the dark at room temperature, cells were washed once more and suspended with Stain Buffer (BD Biosciences). Cytomics FC500 MPL and data analysis software CXP Analysis (both from Beckman Coulter K.K, Shizuoka, Japan) were used for flow cytometry analysis. A gate containing the cell population of interest was created in a forward scatter and side scatter dot plot. Measurement samples were prepared by adding 250 μL of 7-AAD (BD Biosciences; diluted 50-fold in culture medium) to 250 μL of each tumor suspension to detect dead cells. The 7-AAD-positive cell rate (%) and the mean fluorescent intensity of BTK phosphorylation in cells in the relevant gate were measured. Samples in which the dead tumor cell rate (%) was > 30% were not considered suitable for accurate measurement of BTK phosphorylation.

### Gene expression microarray analysis

The TMD8 xenograft model was also used for microarray analysis. When the mean tumor volume in the mouse xenograft model had reached about 150 mm³, tirabrutinib was administered to each mouse twice daily. After 21 days of drug administration and 24 h after the final dose, the mice underwent laparotomy under isoflurane anesthesia and were sacrificed by exsanguination. Tumors were collected and quickly frozen with liquid nitrogen. RNA extraction and gene expression microarray analysis were performed at LSI Medience Corporation (Tokyo, Japan) using the miRNeasy® mini kit (Qiagen, Hilden, Germany) and the Low Input Quick Amp Labeling Kit (Agilent Technologies, Santa Clara, CA, USA) according to the

manufacturers' instructions. The DNA microarray results were processed using Expressionist Analyst (GeneData, Basel, Switzerland) and Excel. The microarray data are accessible through the Gene Expression Omnibus (GEO; GSE 210284).

Gene set enrichment analyses (GSEA) were performed using GSEA version 4.1.0 [34] with hallmark [35], C2, C3, C5, C6, C7, and C8 signatures in mSigDB version 7.4 [36]. The samples from tirabrutinib 10 mg/kg-administered mice (N = 9) and control mice (N = 8) were compared with the default settings of GSEA, except for the permutation method. As the expression of many genes differed between the two groups and sample-wise permutation has many false-positive results, we used gene-wise permutation.

## Results

### Selectivity profile of tirabrutinib: Results from *in vitro* kinase assays

In the comprehensive evaluation of kinase-binding inhibitory activity conducted against a panel of 442 kinases, only five kinases were inhibited by tirabrutinib at 300 nM to a level greater than 65%: BTK, 97%; TEC, 92.2%; BMX, 89%; HUNK, 89%; and RIPK2, 67% (S2 Table). Compared with the first-generation inhibitor ibrutinib, tirabrutinib and the other second-generation BTK inhibitors (acalabrutinib and zanubrutinib) demonstrated more selective kinase binding (Fig 1).

The results of selectivity evaluations against the selected 13 kinases are shown in Table 1. Among the selected kinases, FYN, LCK, and LYNa are analogous kinases of BTK, and nine tyrosine kinases, BLK, BMX, EGFR, ERBB2, ERBB4, ITK, JAK3, TXK, and TEC, have a cysteine residue in the active site like BTK. CSK was also selected because it is reportedly inhibited by ibrutinib and is responsible for off-target toxicity [37]. All the second-generation inhibitors, including tirabrutinib, were more selective against BTK compared with ibrutinib. Of the 13 kinases evaluated for selectivity, only two kinases, BMX and TEC, were within 10-fold selectivity for tirabrutinib. Tirabrutinib showed more than 700-fold selectivity against FYN and LYNa (kinases located upstream of BTK, which are involved in BCR signaling) and more than 200-fold selectivity against ITK, JAK3, LCK, and FYN (kinases involved in T-cell receptor signaling), indicating extremely low off-target kinase inhibition. Tirabrutinib also showed >3597, 3097, 64, and 162-fold selectivity against EGFR, ERBB2, ERBB4, and CSK, respectively.

### Selectivity of tirabrutinib in *in vitro* cellular systems

The data from the PBMC stimulation assays are reported in Fig 2A. Tirabrutinib inhibited anti-IgM-stimulated B-cell activation in a concentration-dependent manner, with an $IC_{50}$ value of 13.8 nM. Conversely, tirabrutinib did not inhibit anti-CD3/CD28-stimulated T-cell activation even at the highest concentration of 10 μM, indicating that tirabrutinib selectively suppressed the B-cell activation marker (CD69) without contributing to T-cell activation.

Fig 2B shows the results obtained from the BioMAP system. Tirabrutinib showed a concentration-dependent inhibitory effect on the BT system (a model of T-cell-dependent B-cell activation), specifically inhibiting B-cell growth and the production of IgG and various cytokines (i.e., interleukin [IL]-6, IL-17A, and tumor necrosis factor [TNF]α). There was no significant effect of tirabrutinib in other evaluation systems such as the T-cell and fibroblast activation systems, further supporting the evidence that tirabrutinib selectively affects B-cells.

### *In vitro* efficacy of tirabrutinib and comprehensive analysis of downstream signaling by phosphoproteomics in ABC-DLBCL cell lines

Tirabrutinib was shown to inhibit the growth of both TMD8 and U-2932 cells (Fig 3A). For TMD8 cells, the $IC_{50}$ was 3.59 nM, and the maximum growth inhibition rate was 98.6% at 100

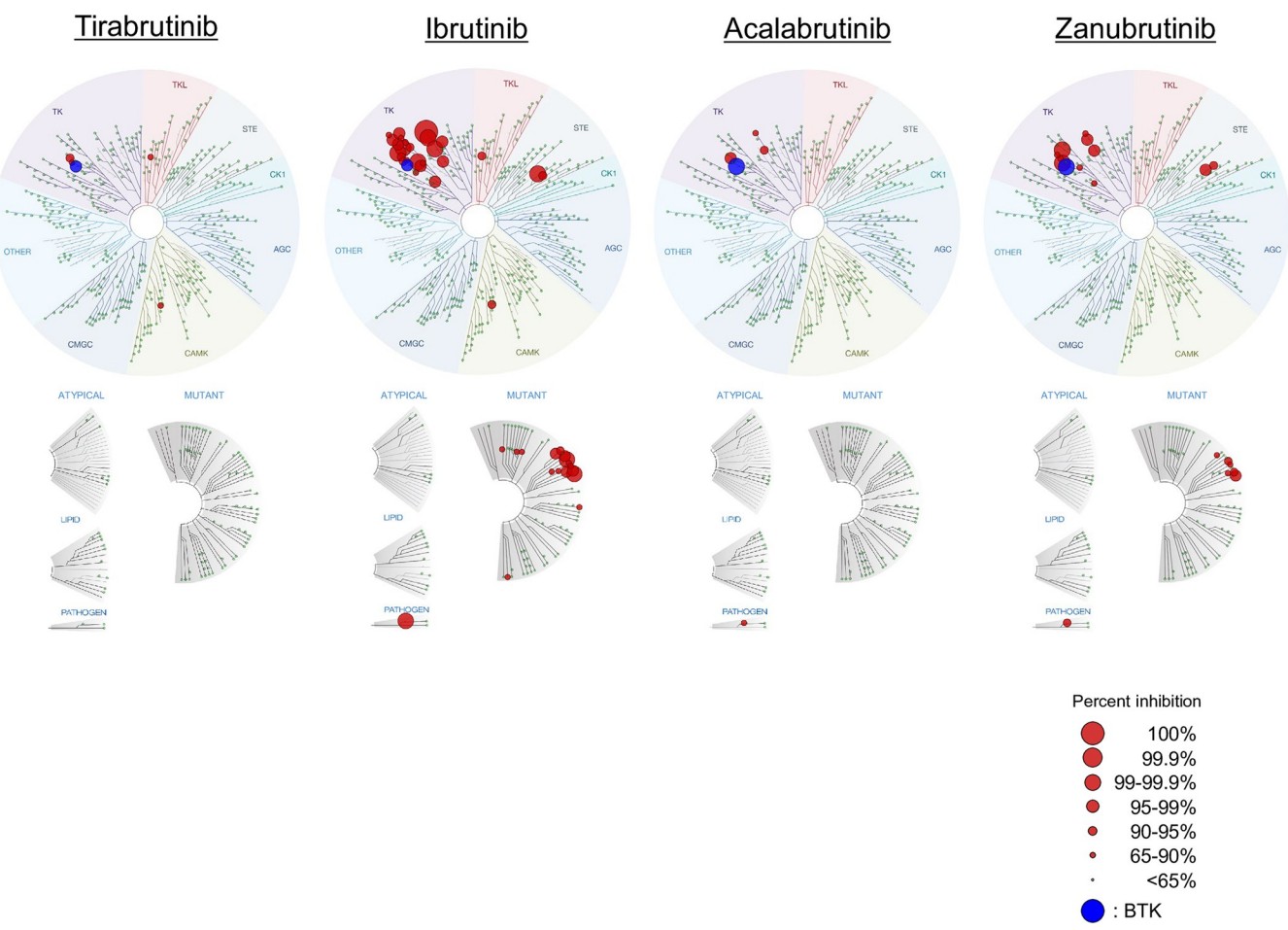

**Fig 1. Kinome profiling using the KINOMEscan assay for the first- and second-generation BTK inhibitors.** KINOMEscan profile against 442 kinases at 300 nM are shown as a TreeSpot diagram, which depicts all test kinases on a circular dendrogram of the human kinome, with interacting kinases (percent inhibition; >65%) shown in blue (BTK) or red (kinases other than BTK); non-interacting kinases are represented as small green dots. The images were generated using TREEspot™ Software Tool and reprinted with permission from KINOMEscan™, a division of Eurofins Corporation (© DISCOVERX CORPORATION 2010). BTK, Bruton's tyrosine kinase.

nM. For U-2932 cells, the $IC_{50}$ was 27.6 nM, and the maximum growth inhibition rate was 30.8% at 100 nM. Tirabrutinib inhibited BTK autophosphorylation at Tyr-223 in both cells in a concentration-dependent manner (**Fig 3B**), with the $IC_{50}$ values (23.9 nM for TMD8 and 12.0 nM for U-2932) being comparable to the $IC_{50}$ for growth inhibition. Thus, proliferation appears to be inhibited by tirabrutinib via inhibition of BTK phosphorylation.

Next, we conducted a phosphoproteomic analysis to investigate phosphoregulation induced by 1-h treatment with tirabrutinib in TMD8 and U-2932 cells, which would result in anti-tumor efficacy. In TMD8 cells, 6,147 distinct phosphoproteins were identified, within which 11,445 distinct phosphorylation sites could be localized to specific serine, threonine, or tyrosine residues with high confidence ($P \geq 0.75$) and were thus classified as class I phosphosites. The overall abundances of Ser(P), Thr(P), and Tyr(P) were 84.6%, 14.4%, and 1.0%, respectively. In replicate analyses, 9,497 distinct phosphosites were identified and quantified, representing a large overlap between the 10,460 and 10,482 phosphorylation sites assigned in the individual experiments. In U-2932 cells, more than 5,563 distinct phosphoproteins and 11,493 distinct phosphorylation sites (class I phosphosites) were identified with high confidence

**Table 1. Selectivity of first-and second-generation BTK inhibitors against 13 kinases.**

| | Tirabrutinib | | Ibrutinib | | Acalabrutinib | | Zanubrutinib | |
|---|---|---|---|---|---|---|---|---|
| | IC$_{50}$ (nM) | Selectivity | IC$_{50}$ (nM) | Selectivity | IC$_{50}$ (nM) | Selectivity | IC$_{50}$ (nM) | Selectivity |
| BTK[a] | 2.78 | 1 | 0.256 | 1 | 4.95 | 1 | 0.285 | 1 |
| FYN | 2220 | 799 | 55.0 | 215 | >10000 | >2020 | 1659 | 5821 |
| LYNa | 3490 | 1255 | 17.8 | 70 | >10000 | >2020 | 734 | 2575 |
| LCK | 788 | 283 | 5.87 | 23 | 5204 | 1051 | 369 | 1293 |
| BLK | 1280 | 460 | 0.155 | 0.6 | 2270 | 459 | 1.71 | 6 |
| BMX | 3.16 | 1 | 0.747 | 3 | 45.1 | 9 | 1.26 | 4 |
| EGFR | >10000 | >3597 | 1.71 | 7 | >10000 | >2020 | 10.7 | 38 |
| ERBB2 | 8610 | 3097 | 3.01 | 12 | 370 | 75 | 34.8 | 122 |
| ERBB4 | 177 | 64 | 0.325 | 1 | 30.1 | 6 | 2.63 | 9 |
| ITK | >10000 | >3597 | 21.9 | 86 | >10000 | >2020 | 346 | 1214 |
| JAK3 | >10000 | >3597 | 14.5 | 57 | >10000 | >2020 | 811 | 2846 |
| TXK | 54.5 | 20 | 4.89 | 19 | 273 | 55 | 4.59 | 16 |
| TEC | 9.92 | 4 | 1.37 | 5 | 13.9 | 3 | 4.47 | 16 |
| CSK | 449 | 162 | 10.7 | 42 | 6653 | 1344 | 194 | 680 |

[a]BTK was used as the reference kinase for calculating the selectivity of the other 13 kinases evaluated.

IC$_{50}$, Half-maximal inhibitory concentration.

($P \geq 0.75$). The overall abundances of Ser(P), Thr(P), and Tyr(P) were 84.6%, 14.8%, and 0.7%, respectively. In replicate analyses, 9,740 distinct phosphosites were identified and quantified, again representing a large overlap between the individual experiments (10,449 and 10,784 sites).

**Fig 4** shows the number of upregulated and downregulated phosphorylation sites in TMD8 and U-2932 cells according to the applied ratio threshold criteria. A total of 138 phosphorylation sites, derived from 125 distinct phosphoproteins, were found to be downregulated in TMD8 cells (1.5% of all quantified sites) (**S3 Table**). The upregulated fraction consisted of 15 sites from 15 phosphoproteins (0.2% of all quantified sites). In U-2932 cells, only 30 phosphorylation sites from 28 distinct phosphoproteins were suppressed (0.3% of all quantified sites) and four phosphorylation sites from four phosphoproteins were induced (0.04% of all quantified sites) (**S4 Table**). Regarding the downregulated phosphorylation sites, five site-specific phosphorylation events that underwent significant regulation in both cell lines were identified (**Table 2**). For the upregulated phosphorylation sites, there was no overlap in TMD8 and U-2932.

Among the five common significantly downregulated phosphosites, ERK2 activation loop phosphorylation on Tyr-187 was suppressed to comparable levels in both cell lines. The phosphokinase inositol 1,4,5-trisphosphate 3-kinase B, which acts as a feedback inhibitor of BCR-induced store-operated Ca$^{2+}$ entry [38], underwent site-specific downregulation on Ser-43, although the significance of this phosphosite is unclear. A GPCR kinase, beta-adrenergic receptor kinase 1 (GRK2) was also found to be downregulated on Ser-685. This phosphosite is reported to be phosphorylated by messenger-governed kinases, such as PKA and PKC, which is involved in desensitization and attenuation of G protein coupling [39,40]. Ser-50 of Golgi vesicular membrane-trafficking protein p18 and Ser-158 of vesicle-associated membrane protein-associated protein B/C were downregulated by tirabrutinib, although there are no reports to describe the function of these phosphosites.

Downregulated phosphosites, specifically in the TMD8 cell lines, included kinases, transcription factors, and other proteins, such as AKT substrates (**Table 3**). Regarding kinases, in

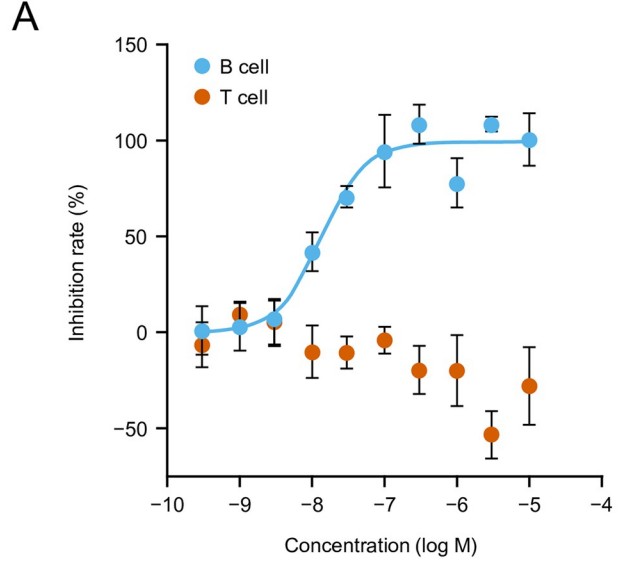

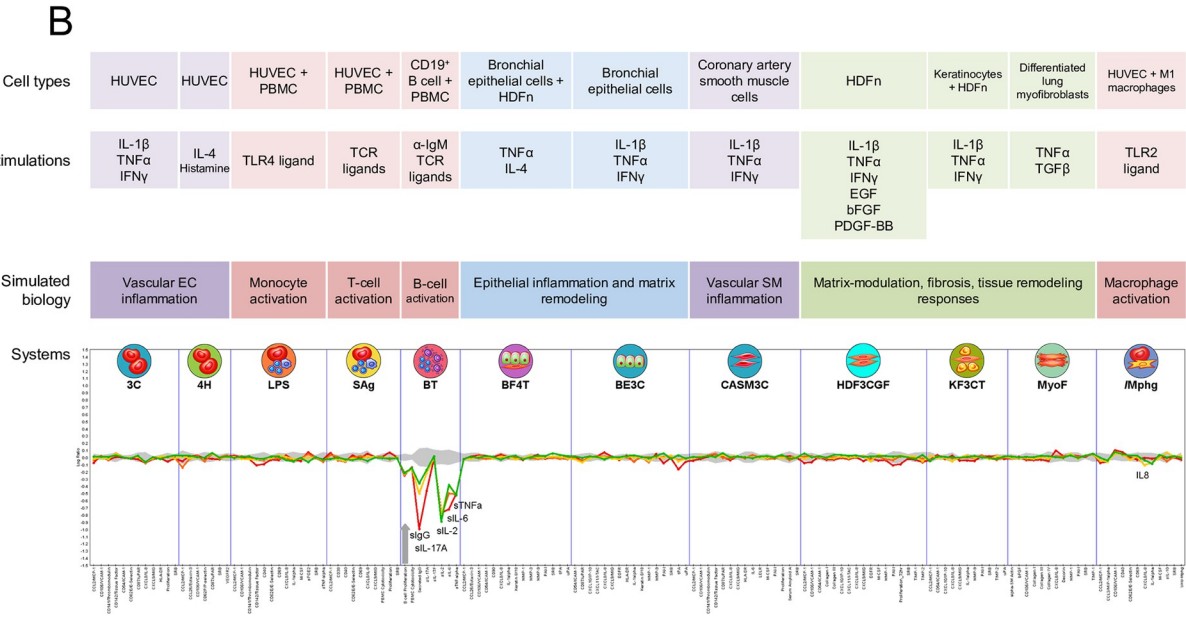

**Fig 2. Selectivity of tirabrutinib in *in vitro* cellular systems.** (A) PBMC activation assays. (B) BioMAP evaluation. (A) Effect of tirabrutinib on B and T cells activation. Human peripheral blood mononuclear cells were treated with vehicle or tirabrutinib, stimulated with anti-IgM or anti-CD3/CD28 antibodies, and labelled with fluorescent anti-CD3, anti-CD20, and anti-CD69 antibodies. After separation by flow cytometry, mean fluorescence intensity of CD69 in each cell population was measured. Values are the mean inhibition rate of three samples ± standard error. (B) The X-axis lists the quantitative protein-based biomarker readouts measured in each system. The Y-axis represents a log-transformed ratio of the biomarker readouts for the tirabrutinib-treated sample (N = 1) over vehicle controls (N ≥ 6). The grey region around the Y-axis represents the 95% significance envelope generated from historical vehicle controls. Biomarker activities are annotated when two or more consecutive concentrations change in the same direction relative to vehicle controls, are outside the significance envelope, and have at least one concentration with an effect size >20% (|log10 ratio| > 0.1). The thick grey arrow denotes antiproliferative effects, which only require one concentration to meet the indicated threshold for profile annotation. bFGF, basic fibroblast growth factor; EGF, epidermal growth factor; HDFn, human neonatal dermal fibroblasts; HUVEC, human umbilical vein endothelial cells; IFN, interferon; Ig, immunoglobulin; IL, interleukin; PBMC, peripheral blood mononuclear cells; PDGF, platelet-derived growth factor; TCR, T-cell receptor; TGF, transforming growth factor; TLR, Toll-like receptor; TNF, tumor necrosis factor.

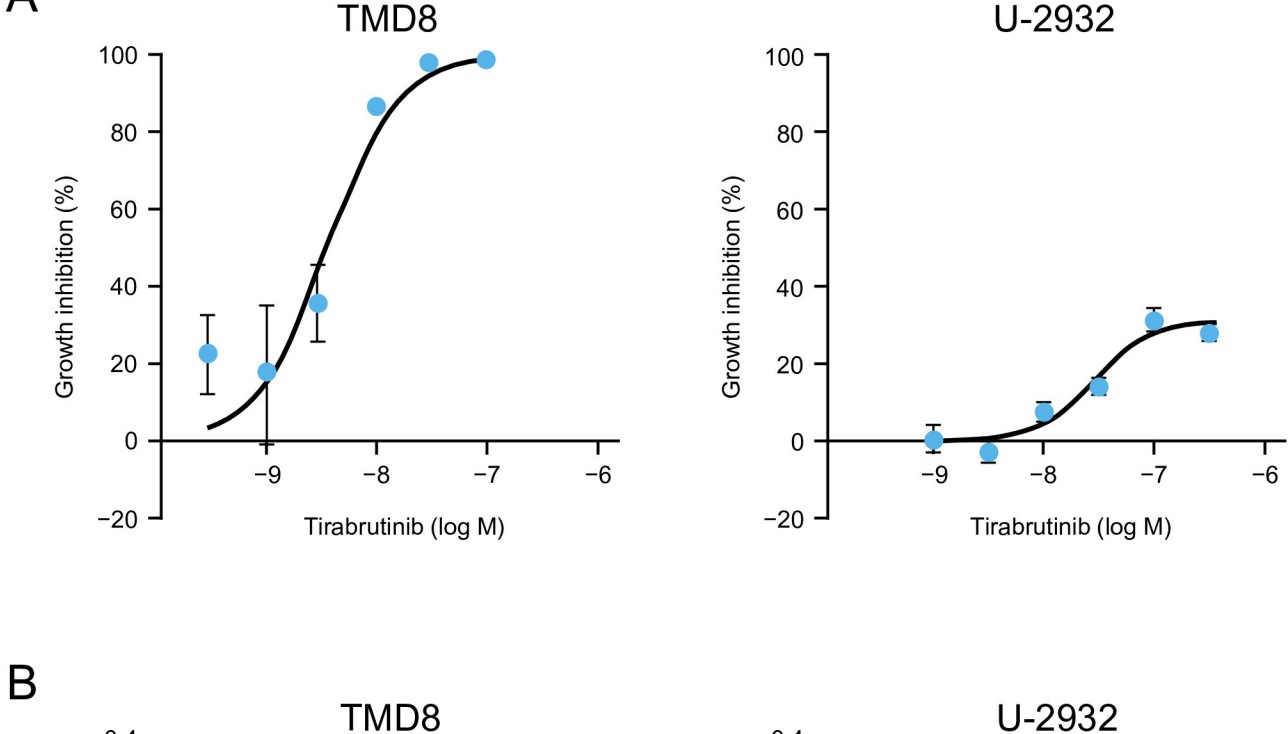

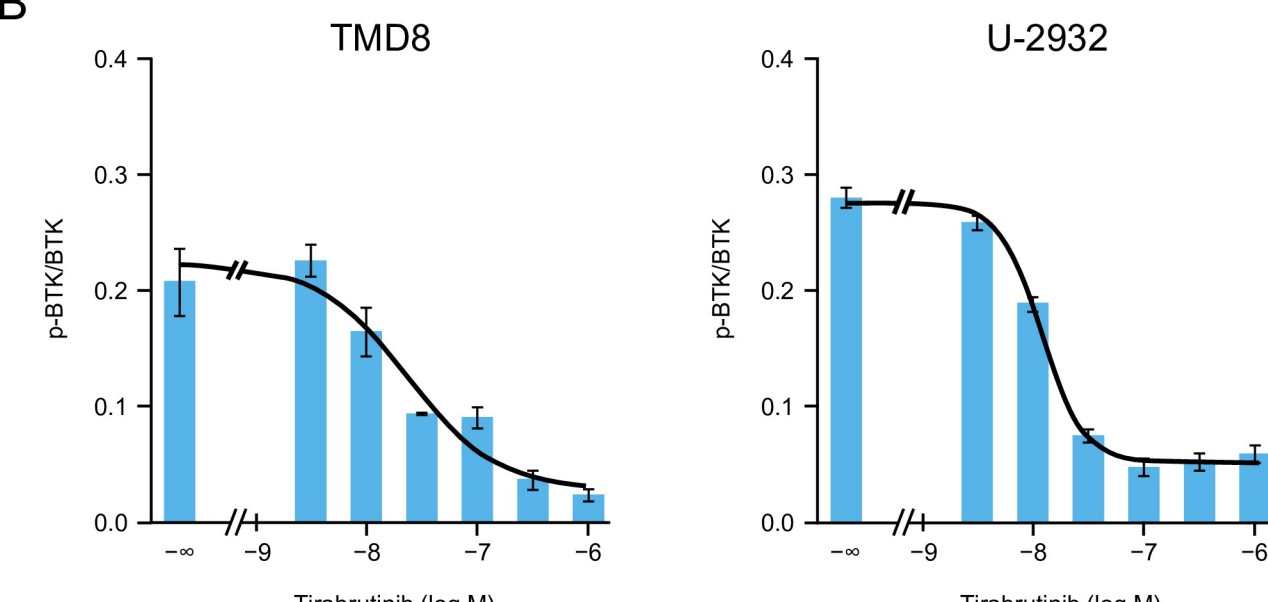

**Fig 3.** Antiproliferative activity (A) and BTK autophosphorylation inhibitory effects (B) of tirabrutinib. (A) TMD8 or U-2932 cells were treated with vehicle or different concentrations of tirabrutinib and incubated for 72 h at 37˚C, 5% $CO_2$/95% air. After culturing, the growth inhibition rate (%) was calculated by measuring a luminescent signal proportional to the amount of intracellular ATP using the CellTiter-Glo Luminescent Cell Viability Assay. The growth inhibition rate in the tirabrutinib group is plotted as the mean of 3 or 4 treated cultures from each treatment group ± standard error for TMD8 or U-2932, respectively. (B) TMD8 or U-2932 cells were treated with vehicle or different concentrations of tirabrutinib and incubated for 4 h at 37˚C, 5% $CO_2$/95% air. Autophosphorylated BTK (p-BTK) and total BTK (BTK) proteins were detected by western blot analysis. The intensity of each western blot band shown in **S1 Fig** was determined. The ratio of phosphorylated to total BTK is described as the mean of three cases ± standard error in the bar chart. The symbol -∞ indicates the vehicle group. The curve was estimated by nonlinear regression analysis using a four-parameter logistic model. BTK, Bruton's tyrosine kinase.

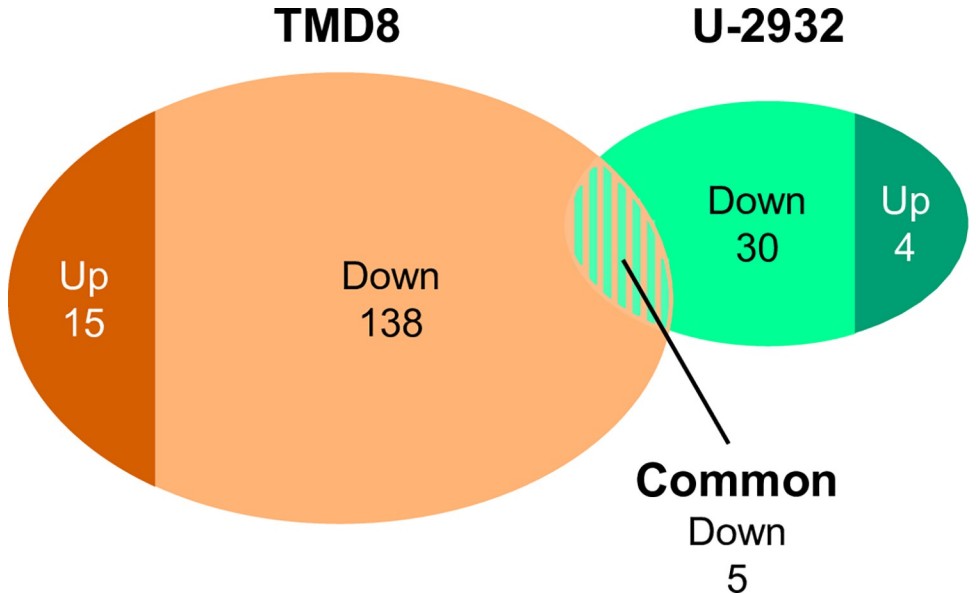

**Fig 4. Number of upregulated or downregulated phosphorylation sites in TMD8 and U-2932 cells.**

addition to ERK1 activation loop phosphorylation on Tyr-204, two sites in the MAPK pathway-associated enzymes Raf-1 and ribosomal S6 kinase 1 (RSK1) were repressed upon tirabrutinib treatment on Ser-1055 and Ser-741, respectively. Although the responsible kinase for C-terminal Ser-741 phosphorylation on RSK1 is not yet known, Raf-1 phosphorylation on Ser-1055 (according to the protein isoform assignment in our database search, identical with Ser-642 in the UniProtKB entry P04049) is known to be directly phosphorylated by ERK kinases [41]. We also detected reproducible inhibition of phosphosites on the related protein kinases PKD2 and PKD3/PKCν, including repression of Ser-710, which is known to lead to catalytic activity of PKD2 and Ser-735, which in turn, is reported to coincide with the kinase activation of PKD3/PKCν [42,43].

Regarding transcription factors, activating transcription factor 2 (ATF2) phosphorylation was reduced by approximately 50% on the adjacent residues Thr-69 and Thr-71, which have been reported to activate transcriptional factor activity of ATF2 [44]. Thr-71 phosphorylation has been reported downstream of the Ras-MEK-ERK pathway [45]. We also detected reduced phosphorylation of Ser-501 of the erythroblast transformation-specific (ETS) domain-containing protein Elk-1 (corresponds to Ser-422 in the UniProtKB entry P19419), a member of the

**Table 2. Significantly downregulated phosphosites in both TMD8 and U-2932 cells.**

| Protein | Gene | Position | Sequence window | Tirabrutinib/ctrl. TMD8 rep. 1 | Tirabrutinib/ctrl. TMD8 rep. 2 | Tirabrutinib/ctrl. U-2932 rep. 1 | Tirabrutinib/ctrl. U-2932 rep. 2 |
|---|---|---|---|---|---|---|---|
| Inositol 1,4,5-trisphosphate 3-kinase B | *ITPKB* | S43 | PRRAVLSPGSVFS | 0.41 | 0.48 | 0.57 | 0.58 |
| Golgi vesicular membrane-trafficking protein p18 | *BET1* | S50 | TAIKSLSIEIGHE | 0.29 | 0.50 | 0.57 | 0.61 |
| Vesicle-associated membrane protein-associated protein B/C | *VAPB* | S158 | IVSKSLSSSLDDT | 0.44 | 0.54 | 0.66 | 0.60 |
| Extracellular signal-regulated kinase 2 | *MAPK1* | Y187 | TGFLTEYVATRWY | 0.34 | 0.47 | 0.23 | 0.41 |
| Beta-adrenergic receptor kinase 1 | *ADRBK1* | S685 | PLVQRGSANGL_ | 0.54 | 0.38 | 0.61 | 0.64 |

**Table 3. Significantly downregulated phosphosites in TMD8 cells.**

| Protein | Gene | Position | Sequence window | Tirabrutinib/ctrl. TMD8 rep. 1 | Tirabrutinib/ctrl. TMD8 rep. 2 | Tirabrutinib/ctrl. U-2932 rep. 1 | Tirabrutinib/ctrl. U-2932 rep. 2 |
|---|---|---|---|---|---|---|---|
| Extracellular signal-regulated kinase 1 | MAPK3 | Y204 | TGFLTEYVATRWY | 0.26 | 0.42 | - | - |
| Proto-oncogene c-RAF kinase | RAF1 | S1055 | ACTLTTSPRLPVF | 0.61 | 0.61 | - | - |
| 90 kDa ribosomal protein S6 kinase 1 (RSK1) | RPS6KA1 | S741 | RVRKLPSTTL_ | 0.61 | 0.59 | 1.16 | 0.87 |
| Serine/threonine-protein kinase D2 (PKD2) | PRKD2 | S214 | RLGTSESLPCTAE | 0.46 | 0.48 | - | - |
| Serine/threonine-protein kinase D2 (PKD2) | PRKD2 | S206 | SLASGHSVRLGTS | 0.32 | 0.48 | 0.86 | 0.83 |
| Serine/threonine-protein kinase D2 (PKD2) | PRKD2 | S710 | EKSFRRSVVGTPA | 0.45 | 0.56 | 0.90 | 0.77 |
| PKCv (PKD3) | PRKD3 | S735 | EKSFRRSVVGTPA | 0.41 | 0.43 | 0.85 | 0.77 |
| Serine/threonine-protein kinase TAO1 | TAOK1 | S9 | STNRAGSLKDPEI | 0.65 | 0.43 | 1.06 | 0.57 |
| Lymphocyte-oriented kinase | LOK | S13 | RRILRLSTFEKRK | 0.48 | 0.60 | 0.85 | 0.77 |
| Sugen kinase 223 | SGK223 | S889 | EKAFKGSGHWLPA | 0.48 | 0.47 | - | - |
| Serine/threonine-protein kinase QSK | SIK3 | S808 | PLSKQLSADSAEA | 0.47 | 0.55 | 1.07 | 0.97 |
| Receptor-interacting serine/threonine-protein kinase 2 | RIPK2 | S531 | VVSRSPSLNLLQN | 0.54 | 0.59 | 1.03 | 0.75 |
| Activating transcription factor 2 | ATF2 | T69 | VIVADQTPTPTRF | 0.53 | 0.53 | - | - |
| Activating transcription factor 2 | ATF2 | T71 | VADQTPTPTRFLK | 0.50 | 0.53 | 0.99 | 0.86 |
| ETS domain-containing protein Elk-1 | ELK1 | S501 | STPVVLSPGPQKP | 0.33 | 0.48 | - | - |
| ETS domain-containing transcription factor ERF | ERF | S327 | VYNYHLSPRAFLH | 0.59 | 0.50 | - | - |
| ETS translocation variant 3 | ETV3 | S245 | MYPDPHSPFAVSP | 0.30 | 0.28 | - | - |
| ETS translocation variant 3 | ETV3 | S250 | HSPFAVSPIPGRG | 0.30 | 0.28 | - | - |
| Myb-like protein 2 | BMYB | T487 | QKVVVTTPLHRDK | 0.67 | 0.50 | - | - |
| Transcription factor RelB | RELB | S37 | ELGALGSPDLSSL | 0.68 | 0.59 | 0.93 | 0.87 |
| Nuclear factor of activated T-cells, cytoplasmic 2 | NFATC2 | S759 | QRSKSLSPSLLGY | 0.67 | 0.61 | 1.14 | 1.19 |
| Nuclear factor of activated T-cells, cytoplasmic 2 | NFATC2 | S243 | PVPRPASRSSSPG | 0.65 | 0.59 | - | - |
| Mitogen-activated protein kinase kinase kinase 7-interacting protein 2 (TAB2) | TAB2 | S450 | GNNSATSPRVVVT | 0.61 | 0.66 | 1.05 | 0.98 |
| L-plastin | LCP1 | S5 | _MARGSVSDEEM | 0.52 | 0.52 | 0.90 | 0.88 |
| Nuclear cap-binding protein subunit 1 | NCBP | S22 | HKRRKTSDANETE | 0.49 | 0.50 | - | 1.04 |
| B-cell adapter for phosphoinositide 3-kinase | BCAP | Y570 | ERPGNFYVSSESI | 0.51 | 0.41 | - | - |
| Astrocytic phosphoprotein PEA-15 | PEA15 | S137 | DIIRQPSEEEIIK | 0.35 | 0.29 | 1.18 | 1.03 |
| Sodium/hydrogen exchanger 1 | NHE1 | S703 | SRARIGSDPLAYE | 0.55 | 0.51 | - | - |
| Heat shock 27 kDa protein | HSP27 | S82 | ALSRQLSSGVSEI | 0.62 | 0.60 | 1.09 | 0.76 |
| Nuclease-sensitive element-binding protein 1 (YB-1) | YBX1 | S102 | PRKYLRSVGDGET | 0.60 | 0.46 | - | - |
| 40 kDa proline-rich AKT substrate (PRAS40) | AKT1S1 | T246 | PRPRLNTSDFQKL | 0.61 | 0.51 | 0.84 | 0.87 |

ETS family. This residue is known to be phosphorylated by ERK1 and involves ternary complex formation, which is a prerequisite for manifesting transactivation potential [46]. As for other transcription factors, site-specific downregulation of phosphosites was detected on Myb-like protein 2 (BMYB) (involved in cell cycle progression [47]), transcription factors as diverse as RelB (a member of the nuclear factor-κB [NF-κB] complex in alternative NF-κB pathway [48]), and nuclear factor of activated T-cells cytoplasmic 2 (NFATC2), although the functions of all the regulated phosphosites by tirabrutinib are unknown.

Other proteins with significantly downregulated phosphosites other than kinases and transcriptional factors were also detected. Tirabrutinib-induced suppression of Ser-450 phosphorylation on MAPKK kinase 7-interacting protein 2 (TAB2), an adaptor protein linked to NF-κB control by the serine/threonine kinase MAP3K7 (TAK1) [49], although the significance of this site is unclear. Tyr-570 phosphorylation on the protein B-cell adapter for phosphoinositide 3-kinase (BCAP) was also suppressed by tirabrutinib. BCAP is reported to play a role in connecting B-cell antigen receptor signaling to phosphoinositide 3-kinase activation through tyrosine phosphorylation [50]. In addition, we identified five distinct AKT substrate proteins, which were all downregulated by tirabrutinib: phosphoprotein enriched in astrocytes 15 (PEA-15), sodium-hydrogen exchanger 1, heat shock protein 27 (Hsp27), nuclease-sensitive element-binding protein 1 (YB-1), and 40-kDa proline-rich AKT substrate (PRAS40). All the identified downregulated phosphosites are reported to be phosphorylated by AKT, while the contribution of AKT is questionable for Ser-703 in sodium-hydrogen exchanger 1 [51–56].

To examine the difference between TMD8 and U2932 in the regulation of BTK downstream signaling by tirabrutinib, PLCγ2, a reported BTK substrate, and other reported or novel direct/indirect downstream effectors in the phosphoproteomic data were sorted (**S5 Table**) and analyzed. Significant suppression was not observed in all the phosphosites of PLCγ2 in either cell line in the phosphoproteomic analysis. In western blot analysis, phosphorylation of PLCγ2 at Tyr-759, which is reported to be phosphorylated by BTK and correlates with the lipase activity of PLCγ2 [57], was inhibited after 1 h of treatment with tirabrutinib at a similar concentration range as that required for the suppression of BTK autophosphorylation in both cell lines (**S2 Fig**). ERK and AKT signaling immunoblot data were consistent with phosphoproteomic data in both TMD8 and U-2932 cells (**S2 and S3 Figs**). Regarding NF-κB signaling, all the phosphorylated sites in the related effectors, including IKKβ and p65, were not downregulated by tirabrutinib in either cell line in the phosphoproteomic analysis. In western blot analysis, however, p-IκBα at S32/36, which is critical for the activation of NF-κB signaling [58], was inhibited after treatment with tirabrutinib for 4 h in both cell lines (**S3 Fig**).

### *In vivo* efficacy of tirabrutinib and comprehensive analysis of downstream signaling by microarray in TMD8

In the TMD8 subcutaneous xenograft transplantation model, tirabrutinib showed a dose-dependent anti-tumor effect. Three tirabrutinib doses (1, 3, and 10 mg/kg BID) were evaluated for anti-tumor effect, and dose-dependent inhibition was observed (**Fig 5A**). After repeated administration of the same three doses of tirabrutinib, inhibition of BTK phosphorylation in tumors was evaluated 1 h after the last dose, indicating a dose-dependent reduction in BTK phosphorylation (**Fig 5B**). Taken together, these results indicate that in this model, tirabrutinib exerted anti-tumor effects in response to BTK inhibition.

To analyze the mechanism of the anti-tumor effect of tirabrutinib in this model, gene expression in the tumor was comprehensively analyzed by microarray after repeated administration of tirabrutinib at 3 or 10 mg/kg BID for 21 days. Differentially expressed gene analysis between vehicle and tirabrutinib 10 mg/kg groups showed that 575 probes were

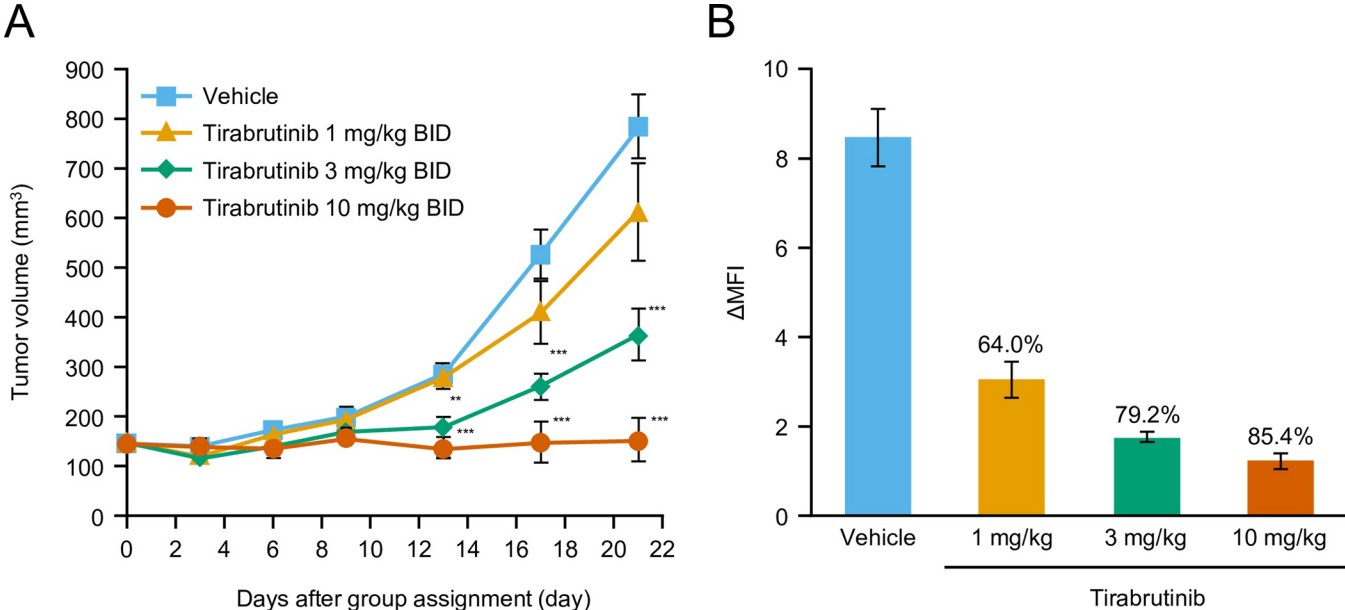

**Fig 5. *In vivo* analyses using mouse xenograft models.** (A) Anti-tumor effect of tirabrutinib in a mouse xenograft model. (B) Effect of tirabrutinib on BTK phosphorylation in the mouse xenograft model. A TMD8 cell suspension (0.1 mL, $1 \times 10^8$ cells/mL) was subcutaneously implanted in SCID mice under pentobarbital anesthesia. Animals were randomized into groups based on tumor volume calculated from the measurement of tumor diameter on the day after implantation. Tirabrutinib or 0.5% methyl cellulose was orally administered twice daily for 21 days, starting on day 0 (the group assignment day). Tumor diameter was measured, and tumor volume was calculated every 3 or 4 days after group assignment. (A) Tumor volume in the figure is expressed as the mean ± standard error in the 10 animals in each group on each measurement day. Dunnett test was used to compare tumor volume in the vehicle- and tirabrutinib-treated groups. A *P*-value of less than 5% was considered statistically significant. **: $P < 0.01$, ***: $P < 0.001$. Results of linear regression analysis in the tirabrutinib-treated group on day 21: The *P*-value of the slope was 0.0001. (B) Tumors were collected 1 h after the final administration. The mean fluorescence intensity (MFI) of BTK phosphorylation in each tumor cell with or without stimulation with $H_2O_2$ was determined. The difference in MFI between the $H_2O_2$-stimulated and unstimulated samples was expressed as the mean ± standard error in 4–8 animals in each group. The BTK phosphorylation inhibition rate (%) in the tirabrutinib-treated groups versus the vehicle group was calculated. BID, twice daily; BTK, Bruton's tyrosine kinase; MFI, mean fluorescent intensity.

downregulated, while 549 probes were upregulated by tirabrutinib treatment (fold change >2 or <0.5, $P < 0.01$) (**S6 Table**). Among the probes that had the highest altered expression (threshold is shown in **Fig 6A**), dose-dependent expression change was observed in tirabrutinib 3 mg/kg and 10 mg/kg groups (**Fig 6B**) consistent with tumor response (**Fig 5A**). C-X-C motif chemokine ligand 10 (CXCL10), which is relevant to lymphoma cell survival [59], was significantly downregulated in the tirabrutinib treatment group (fold change 0.068739, *P*-value = 8.62E-12), whereas arachidonic acid 5-lipoxygenase (ALOX5, fold change 9.19, *P*-value = 6.94E-14) was upregulated, which suggests BCR signal suppression [60].

GSEA was performed to understand molecular mechanisms underlying anti-tumor activity of tirabrutinib in TMD8 tumor (**S7 Table**). Tirabrutinib treatment decreased expression of IRF4 (fold change, 0.8; $P = 0.016$), which is a transcription factor acting downstream of the BCR and NF-κB pathway and the master regulator of B cell differentiation, and genes regulated by IRF4 (**Figs 6C, S4 and S5**), while BCL6, which is a key transcriptional repressor for germinal center (GC) B cell and downregulated by IRF4 activation during the course of B cell development, showed increased expression after BTK inhibition (fold change, 1.2; $P = 0.005$) (**S6 Table**). In addition, it is also suggested that the suppression of MYC and mTORC1 signature contributed to the tumor growth inhibition (**Fig 6C**). To confirm the results by microarray analysis, the changes in mRNA and protein expression levels of IRF4, MYC, and BCL6 after treatment with tirabrutinib for 24 h in TMD8 cells were evaluated by RT-PCR and western blotting. As a result, *IRF4* and *MYC* genes were downregulated by tirabrutinib, whereas

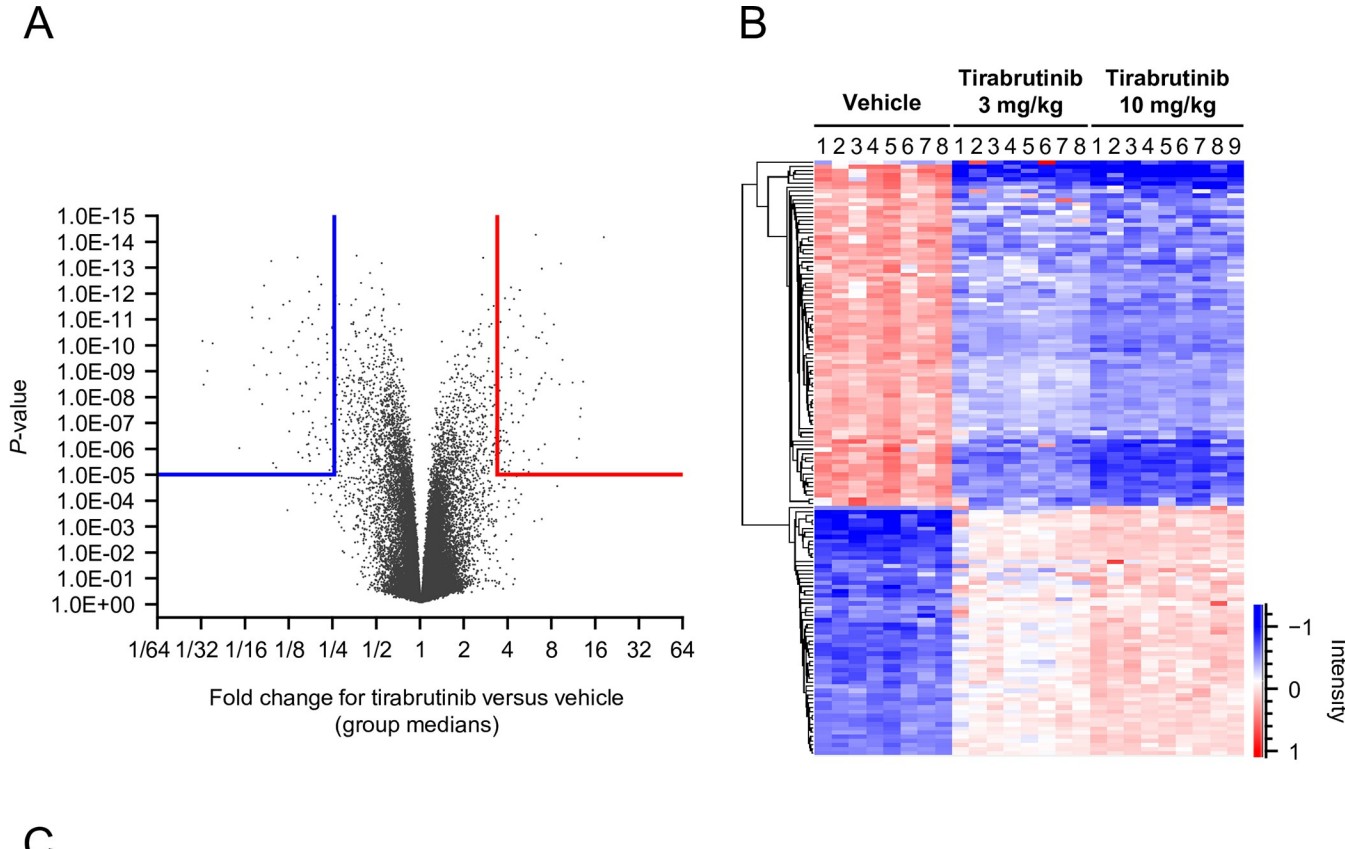

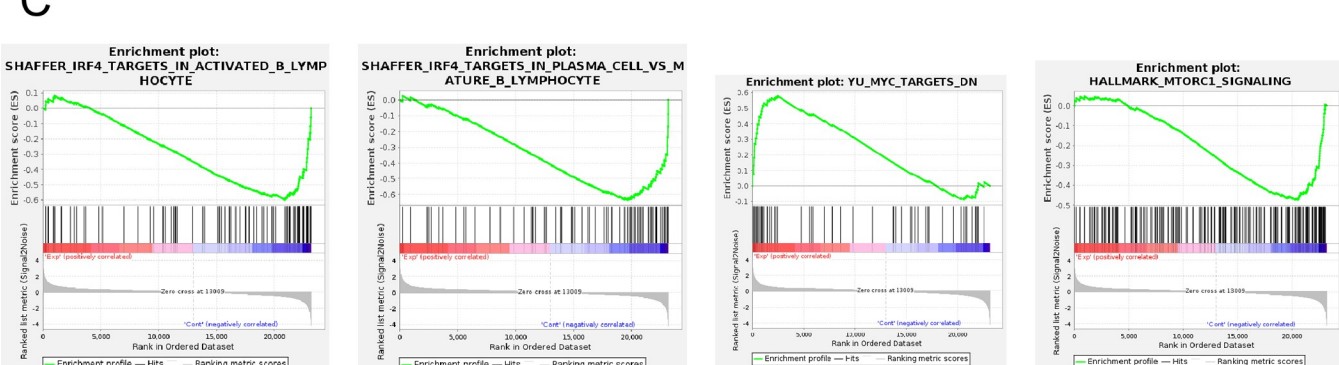

**Fig 6. Global gene expression analysis of response to tirabrutinib treatment in a TMD-8 xenograft model.** Microarray analysis was conducted in the TMD8 xenograft model after 21 days of tirabrutinib administration. (A) Volcano plot for differential gene expression between the vehicle and tirabrutinib 10 mg/kg group with Welch's *t* test. (B) Heat map of top altered gene group including 59 upregulated probes and 87 downregulated probes (fold change, >4 or <0.25; *P* < 0.00001) in the vehicle, tirabrutinib 3 mg/kg, and 10 mg/kg groups. (C) Gene set enrichment analysis (GSEA) plots involved in IRF4, MYC, and mTORC1 signaling in the tirabrutinib 10 mg/kg group versus vehicle group.

the *BCL6* gene was upregulated (**S6 Fig**), which were consistent with the results from microarray. In addition, the downregulation of IRF4 and MYC protein expression by tirabrutinib was observed, although BCL6 protein expression was not changed.

## Discussion

All the second-generation BTK inhibitors, including tirabrutinib, were reported to be more selective against BTK compared with the first generation BTK inhibitor, ibrutinib [14,61]. The

binding of zanubrutinib to BTK is different from that of ibrutinib to BTK, which contributes to its superior selectivity [14]. The binding mode of tirabrutinib and acalabrutinib with BTK is similar to that of ibrutinib, but the structures of the inhibitory component are different. Butylamide electrophiles are less reactive than acrylamide, and the higher nucleophilicity of Cys-481 in BTK may explain the superior selectivity over ibrutinib [13].

Previous research has demonstrated that BCR signaling is chronically active in ABC-DLBCL [20] and that BTK inhibitors are highly effective against ABC-DLBCL malignancies [20,62]. However, comprehensive analysis by both phosphoproteomics and transcriptomics in ABC-DLBCL cell lines to elucidate the anti-tumor mechanism of BTK inhibitors has not been reported previously. Therefore, we examined the mechanism of tirabrutinib in ABC-DLBCL by combining phosphoproteomic and transcriptomic analyses. Before the analysis, we checked off-target kinases inhibited by tirabrutinib using both biochemical and cellular methods in order to understand the possibility of unexpected anti-tumor mechanisms based on the off-target effects of tirabrutinib.

A comprehensive evaluation of kinase-binding inhibitory activity assay demonstrated that only four off-target kinases, namely TEC, BMX, HUNK, and RIPK2, were inhibited by tirabrutinib, while there are no significant reports to indicate the involvement of these four kinases in anti-tumor effects in hematological malignancies. Next, for further investigation, we evaluated the $IC_{50}$ values of tirabrutinib against the selected 13 kinases, including FYN, LCK, and LYNa, which are kinases analogous to BTK; nine tyrosine kinases, including EGFR, ERBB2, ERBB4, ITK, and JAK3, that are similar to BTK in that they have a cysteine residue in the active site; and CSK, which is reported to be inhibited by ibrutinib [37]. Among the 13 kinases examined, there were no kinases that were strongly inhibited by tirabrutinib, except for TEC and BMX.

In addition, our data clearly showed that tirabrutinib did not inhibit T-cell activation or function in the cellular system, and this was consistent with data indicating that tirabrutinib does not inhibit any kinases involved in T-cell function, including ITK, JAK3, or LCK. These data suggest that tirabrutinib is a highly potent and selective BTK inhibitor.

In this analysis, tirabrutinib exerted anti-tumor effects on two ABC-DLBCL cell lines, TMD8 and U-2932, when grown *in vitro* or as a murine xenograft *in vivo*. The growth inhibition of the tumor cells was related to dose-dependent BTK inhibition by tirabrutinib. Unlike TMD8, the maximum growth inhibitory effect of tirabrutinib was limited in U-2932 cells. This finding was not unexpected as it has been reported that the growth inhibitory effect of BTK knockdown by shRNA is limited [20].

*In vitro* comprehensive phosphoproteomic and immunoblotting analyses performed to analyze the mechanism of the anti-tumor effect on ABC-DLBCL suggested that tirabrutinib exerts its effect in both TMD8 and U-2932 cells by downregulating the phosphorylation signal of ERK and NF-κB signaling, which have been reported as downstream signals of BTK [63,64]. Particularly in TMD8 cells, in addition to ERK1/2, downregulated phosphosites through ERK kinases were identified in Elk1and ATF2. Elk1 is reported to have roles in cancer development, including cell proliferation, the cell cycle, apoptosis, and tumorigenesis, although its role in DLBCL remains unclear [65]. ATF2, a member of the AP-1 family, could have an important role in ABC-DLBCL as it has been reported that knockdown of ATF2 by shRNA significantly reduced the survival of an ABC-DLBCL cell line [66]. In addition to ERK and NF-κB phosphorylation signals, downregulation of AKT signals was also indicated only in TMD8 based on the decrease of phosphosites of four distinct AKT substrates (PEA-15, Hsp27, YB-1 and PRAS40) and inhibition of p-AKT at Ser-473 in the immunoblotting analysis. The modification in PEA-15 at Ser-137, which is identical to Ser-116 in the UniProtKB entry Q15121, is known to stabilize the anti-apoptotic action of PEA-15 [51]. Ser-82 phosphorylation of Hsp27 has been reported to control the anti-apoptotic activity of the heat shock protein [52]. It has

been reported that YB-1 could promote the proliferation of DLBCL cells by accelerating the G1/S transition, and expression of phospho-YB-1 (Ser-102) is one of independent predictors of overall survival in DLBCL patients [67]. PRAS40 plays a role in phosphatidylinositol 3-kinase (PI3K)/AKT survival signaling, and Thr-246 phosphorylation has been shown to regulate the functional activity of this protein [68]. AKT signaling-related regulation was also suggested by downregulation of Tyr-570 in BCAP, an adaptor protein connecting B-cell antigen receptor signaling to phosphoinoside 3-kinase activation through tyrosine phosphorylation. Since the tyrosine phosphorylation of BCAP is reported to be mediated by Syk and BTK upon BCR activation [50], the observed downregulation of Tyr-570 in BCAP by tirabrutinib is reasonable as a consequence of BTK inhibition by tirabrutinib. Collectively, inhibition of ERK, AKT, and NF-κB signaling followed by BTK inhibition could contribute as important mechanisms of the anti-tumor effects exerted by tirabrutinib in TMD8. In U-2932 cells, inhibition of both p-ERK and NF-κB signaling was indicated, and the concentration at which tirabrutinib inhibited p-ERK was consistent with that for inhibition of p-BTK and $IC_{50}$ for growth inhibition, indicating that ERK signaling, as a BTK downstream signal, is involved in cell growth. In contrast to TMD8, however, phosphoproteomic and immunoblotting analyses did not indicate inhibition of AKT signaling by tirabrutinib in U-2932 (**S2B Fig**). This may be one of the reasons for the limited anti-tumor effect of tirabrutinib in U-2932, and this finding was consistent with that of a previous study, which demonstrated that the contribution of AKT signaling to survival in U-2932 causes the low sensitivity to BTK inhibitors [69]. The observed significant suppression of PKD2 and PKD3/PKCν by tirabrutinib are other potential phosphoregulations, which may be involved in the anti-tumor effects in TMD8. Given the previous report demonstrating that BCR-induced activation of PKCν is abrogated in a BTK-deficient B cell line, the observed phosphoregulation would be reasonable [70]. Since the role of both PKD2 and PKD3/PKCν in the pathology of DLBCL is unclear, further investigation is needed to clarify whether the observed regulations contribute to the anti-tumor efficacy of tirabrutinib in TMD8.

*In vivo* transcriptome analysis using microarray revealed that tirabrutinib downregulated IRF4, MYC, and mTORC1 pathway-related genes in TMD8 (**Fig 6C**). It is reported that NF-κB signaling is constitutively activated, followed by chronic active BCR signaling in ABC-DLBCL cell lines [20]. NF-κB activates the IRF4 transcription factor through BCR, CD40, and TLR signaling pathways and plays a critical role in B-cell differentiation and the pathogenesis of DLBCL [71]. Downregulation of IRF4 expression by shRNA and lenalidomide selectively killed non-GCB-DLBCL and showed a synergistic effect in combination with ibrutinib [72,73], which indicates that IRF4 is a key regulator in ABC-DLBCL survival. It is also known that there is a positive feedback loop between IRF4 and MYC in B-cell malignancies, which contributes to cell proliferation and survival [74,75]. IRF4 activation leads to BCL6 suppression through the induction of plasma cell differentiation factor BLIMP1 (encoded by PRDM1), which results in MYC overexpression [76]. In accordance with these reports, tirabrutinib treatment repressed PRDM1 and induced BCL6 expression in the TMD8 xenograft model (**S6 Table**). Retrospective clinical studies suggest that ABC-DLBCL patients with MCD genetic subtype (MYD88 and CD79B mutation) showed higher expression of IRF4 and MYC signature, and strong sensitivity to BTK inhibitor with R-CHOP [77,78]. To examine the signaling pathways that regulate gene expression of *IRF4* and *MYC* in TMD8, an IKK inhibitor (BMS-345541), MEK inhibitor (refametinib), and AKT inhibitor (MK-2206) were tested in addition to tirabrutinib. Of these inhibitors, only BMS-345541 and tirabrutinib strongly inhibited *IRF4* expression (**S7 Fig**), suggesting that tirabrutinib inhibits IRF4 expression through NF-κB signaling. As for MYC expression, in addition to strong inhibition by BMS-345541, refametinib and tirabrutinib partially downregulated *MYC*, suggesting that tirabrutinib

downregulated this gene through not only NF-κB but also ERK signaling. Collectively, BTK inhibition by tirabrutinib led to the suppression of NF-κB and ERK signaling, followed by downregulation of IRF4- and MYC-targeted genes, resulting in anti-tumor activity in TMD8. Another interesting finding is that the expression of CXCL10, reported to be highly expressed in lymphoma patients [59], was decreased, while the expression of ALOX5 was increased; this enzyme is suppressed by BCR signaling and has previously been reported to be increased in TMD8 cells by knockdown of CD79B and by ibrutinib [60].

We also observed some correlations between the results of the phosphoproteomic and transcriptomic analyses of signaling pathways regulated by tirabrutinib in TMD8. For example, tirabrutinib significantly decreased the phosphorylation of ATF2 (Thr-71), which is phosphorylated by ERK, leading to transcriptional activation. ATF2 is reported to control ATF3 expression, which is an important element controlling the growth of ABC-DLBCL [66]. In the transcriptomic analysis, the expression of ATF3 was significantly suppressed by tirabrutinib (fold change, 0.36; $P < 0.001$). In addition, the observed downregulation at Thr-246 of PRAS40 by tirabrutinib in the phosphoproteomic analysis suggests the suppression of mTORC1 signaling [79], which is consistent with the downregulation of mTORC1 pathway-related genes by tirabrutinib in the transcriptomic analysis.

## Conclusion

In summary, the highly selective BTK inhibitor tirabrutinib exerted an anti-tumor effect via the regulation of multiple BTK downstream signaling proteins, such as NF-κB, AKT, and ERK, in ABC-DLBCL cell lines.

## Supporting information

**S1 Fig. Immunoblots of lysates to evaluate BTK autophosphorylation inhibitory effects of tirabrutinib in the TMD8 and U-2932 cell lines.** (**A**) TMD8 and (**B**) U-2932 cells were treated with vehicle or different concentrations of tirabrutinib and incubated for 4 h at 37˚C (5% $CO_2$/95% air). Autophosphorylated BTK (p-BTK) (upper) and total BTK (BTK) (lower) proteins were detected by western blot analysis. TMD8 cells were unstimulated (–) or stimulated using $H_2O_2$ and used as a marker for the detection of p-BTK and BTK.
(PDF)

**S2 Fig. Immunoblots for p-BTK, BTK, p-PLCγ2, PLCγ2, p-ERK1/2, ERK1/2, and GAPDH in lysates of TMD8 and U-2932 cells treated with or without tirabrutinib (0–300 nM). (A)** TMD8 cells or (**B**) U-2932 cells were treated with tirabrutinib (10, 30, 100, or 300 nM) or DMSO and incubated for 1 and/or 4 h at 37˚C in 5% $CO_2$/95% air. Autophosphorylated BTK (p-BTK, Tyr-223), total BTK (BTK), phosphorylated PLCγ2 (p-PLCγ2, Tyr-759), total PLCγ2 (PLCγ2), phosphorylated ERK1/2 (p-ERK, Thr-202/204), total ERK1/2, and GAPDH proteins were detected by western blot analysis. TMD8 cells were stimulated using $H_2O_2$ and used as a marker for the detection of p-BTK, p-PLCγ2, and p-ERK.
(PDF)

**S3 Fig. Immunoblots of p-AKT, AKT, p-IκBα, IκBα, and GAPDH in lysates of TMD8 and U-2932 cells treated with or without tirabrutinib (1 μM).** U-2932 cells and TMD8 were treated with tirabrutinib (1 μM) or DMSO and incubated for 1 or 4 h at 37˚C in 5% $CO_2$/95% air. (**A**) Phosphorylated AKT (p-AKT, Ser-473) and total AKT (AKT) proteins were detected by western blot analysis. (**B**) Phosphorylated IκBα (p-IκBα, Ser-32/36), total IκBα (IκBα), and GAPDH proteins were detected by western blot analysis.
(PDF)

**S4 Fig. Heat map of differentially expressed genes in gene set enrichment analysis.** Heat map of four significantly enriched gene sets between vehicle group (N = 8) and tirabrutinib 10 mg/kg group (N = 9). (A) Shaffer_IRF4_targets_in_activated_B_lymphocyte, (B) Shaffer_-IRF4_targets_in_plasma_cell_vs_mature_B_lymphocye, (C) YU_MYC_targets_up, (D) Hall-mark_MTORC1_signaling.
(PDF)

**S5 Fig. Core enrichment genes of gene set enrichment analysis for interferon regulatory factor target genes in the two gene sets.**
(PDF)

**S6 Fig. Gene and protein expression in TMD8 cells treated with or without tirabrutinib.** TMD8 were treated with tirabrutinib (1 μM) or DMSO and incubated for 24 h at 37˚C in 5% $CO_2$/95% air. (**A**) RNA was isolated and polymerase chain reaction (PCR) was performed. All PCRs were performed in triplicate, and the mRNA expression of GAPDH was used as an internal control. The *t* test was used to compare mRNA expression in the DMSO- and tirabrutinib-treated groups. A *P*-value of less than 5% was considered statistically significant. **: $P < 0.01$, ***: $P < 0.001$. (**B**) IRF4, BCL6, MYC, and GAPDH proteins were detected by western blot analysis.
(PDF)

**S7 Fig. Gene expression detected by RT-PCR in TMD8 cells treated with or without various selective inhibitors.** TMD8 were treated with tirabrutinib (1 μM), BMS-345541 (10 μM), MK-2206 (1 μM), refametinib (0.1 μM) or DMSO and incubated for 24 h at 37˚C in 5% $CO_2$/95% air. RNA was isolated and polymerase chain reaction (PCR) was performed. All PCRs were performed in triplicate, and the mRNA expression of GAPDH was used as an internal control. The Dunnett test was used to compare mRNA expression in the DMSO- and compound-treated groups. A *P*-value of less than 5% was considered statistically significant. *: $P < 0.05$, ***: $P < 0.001$. n.s.: Not significant.
(PDF)

**S1 Table. The 12 *in vitro* systems of the BioMAP Diversity PLUS panel (Eurofins Disco-verX Products, LLC).**
(DOCX)

**S2 Table. Complete KINOMEscan dataset.** The data represent percent control (%CTRL) as the percent of each kinase that remains bound to the immobilized ligand. Therefore, a value of 0% control equals 100% blocking of kinase-ligand binding, while a value of 100% equals no binding of each compound to the target kinase.
(XLSX)

**S3 Table. Processed version of the MaxQuant phosphosite table of the experiments with tirabrutinib-treated TMD8 cells.** For all confidently localized phosphorylation sites (localization probability ≥0.75%), SILAC ratios of individual replicate experiments are reported. Class 1 site ratios that were reproducibly identified are indicated and further marked as regulated in case of consistent up- or downregulation. The tirabrutinib/control phosphosite ratios from Experiment 2 correspond to the normalized and inverted H/L ratio due to the introduced label switch compared to Experiment 1.
(XLSX)

**S4 Table. Processed version of the MaxQuant phosphosite table of the experiments with tirabrutinib-treated U-2932 cells.** For all confidently localized phosphorylation sites

(localization probability ≥0.75%), SILAC ratios of individual replicate experiments are reported. Class 1 site ratios that were reproducibly identified are indicated and further marked as regulated in case of consistent up- or downregulation. The tirabrutinib/control phosphosite ratios from Experiment 2 correspond to the normalized and inverted H/L ratio due to the introduced label switch compared to Experiment 1.
(XLSX)

**S5 Table. Sorted phosphosites of BTK-related effectors in both TMD8 and U-2932 cells.** Phosphosites of BTK-related effectors in both TMD8 and U-2932 cells were sorted from S3 Table and S4 Table and the responsible kinase and/or function of each phosphorylation site are described.
(XLSX)

**S6 Table. Comprehensive gene expression analysis using microarray in TMD8 xenograft model.** The microarray data were normalized to the 75[th] percentile signal intensity as recommended by Agilent. Welch's *t*-test of log-converted expression values was performed to find significant differentially expressed genes between the tirabrutinib and vehicle groups. Fold changes, nominal *P*-values and q-values of the differentially expressed genes are shown. Benjamini-Hochberg method was used as the correction method for the q-value. Geometric means of expression values for the tirabrutinib and vehicle groups are also shown.
(XLSX)

**S7 Table. Gene set enrichment analysis between tirabrutinib 10 mg/kg and vehicle in TMD8 xenograft model.** Gene set enrichment analysis (GSEA) identified 710 and 2216 significantly enriched gene sets in the vehicle group and tirabrutinib 10 mg/kg group, respectively, with $P < 0.05$ and FDR q $< 0.25$.
(XLSX)

**S1 Raw images.**
(PDF)

**S2 Raw images.**
(PDF)

**S3 Raw images.**
(PDF)

**S4 Raw images.**
(PDF)

## Acknowledgments

We thank Sally-Anne Mitchell, PhD, of Edanz, Japan, for providing medical writing support through EMC K.K., Japan, in accordance with Good Publication Practice (GPP 2022) guidelines (https://www.ismpp.org/gpp-2022).

## Author Contributions

**Conceptualization:** Ryohei Kozaki.

**Data curation:** Ryohei Kozaki, Tomoko Yasuhiro, Hikaru Kato, Jun Murai, Shingo Hotta, Yuko Ariza.

**Formal analysis:** Ryohei Kozaki, Tomoko Yasuhiro, Hikaru Kato, Jun Murai, Shingo Hotta, Yuko Ariza, Shunsuke Sakai, Ryu Fujikawa.

**Investigation:** Ryohei Kozaki, Tomoko Yasuhiro, Hikaru Kato, Jun Murai, Shingo Hotta, Yuko Ariza, Shunsuke Sakai, Ryu Fujikawa.

**Methodology:** Ryohei Kozaki, Tomoko Yasuhiro, Hikaru Kato, Jun Murai, Shingo Hotta, Yuko Ariza, Shunsuke Sakai, Ryu Fujikawa.

**Project administration:** Ryohei Kozaki, Tomoko Yasuhiro, Hikaru Kato, Jun Murai, Shingo Hotta, Yuko Ariza.

**Resources:** Ryohei Kozaki, Tomoko Yasuhiro, Shingo Hotta, Yuko Ariza.

**Supervision:** Takao Yoshida.

**Validation:** Ryohei Kozaki, Tomoko Yasuhiro, Hikaru Kato, Jun Murai, Shingo Hotta, Yuko Ariza.

**Visualization:** Ryohei Kozaki, Tomoko Yasuhiro, Hikaru Kato, Jun Murai, Shingo Hotta, Yuko Ariza.

**Writing – original draft:** Ryohei Kozaki, Tomoko Yasuhiro, Hikaru Kato, Jun Murai, Shingo Hotta, Yuko Ariza, Takao Yoshida.

**Writing – review & editing:** Ryohei Kozaki, Tomoko Yasuhiro, Hikaru Kato, Jun Murai, Shingo Hotta, Yuko Ariza, Shunsuke Sakai, Ryu Fujikawa, Takao Yoshida.

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
