## [Decision Letter · Decision Letter 0]

5 Jul 2022

PONE-D-22-16384Investigation of the anti-tumor mechanism of tirabrutinib, a highly selective Bruton’s tyrosine kinase inhibitor, by phosphoproteomics and transcriptomicsPLOS ONE

Dear Dr. Kozaki,

Thank you for submitting your manuscript to PLOS ONE. After careful consideration, we feel that it has merit but does not fully meet PLOS ONE’s publication criteria as it currently stands. Therefore, we invite you to submit a revised version of the manuscript that addresses the points raised during the review process.

We look forward to receiving your revised manuscript.

Kind regards,

Mauro Salvi, Ph.D.

Academic Editor

PLOS ONE

Journal Requirements:

"I have read the journal's policy and the authors of this manuscript have the following competing interests: This study was funded by Ono Pharmaceutical Co., Ltd. (Osaka, Japan). All authors are employees of Ono Pharmaceutical Co., Ltd."  

We note that you received funding from a commercial source: "Ono Pharmaceutical Co., Ltd."

Additional Editor Comments:

The manuscript has been reviewed by two independent reviewers and all agree that it is a work worth of publication, but it requires some amendments as suggested by the Reviewers.

Therefore, we invite you to submit a revised version of the manuscript that addresses the points raised during the review process. In particular Reviewer 1 requires some experiments to confirm the mass spectrometry and/or the transcriptomic analysis.

Reviewers' comments:

Reviewer's Responses to Questions

**Comments to the Author**

1. Is the manuscript technically sound, and do the data support the conclusions?

Reviewer #1: Partly

Reviewer #2: Yes

2. Has the statistical analysis been performed appropriately and rigorously? 

Reviewer #1: Yes

Reviewer #2: Yes

3. Have the authors made all data underlying the findings in their manuscript fully available?

Reviewer #1: Yes

Reviewer #2: Yes

4. Is the manuscript presented in an intelligible fashion and written in standard English?

Reviewer #1: Yes

Reviewer #2: Yes

5. Review Comments to the Author

Reviewer #1: In this work, Kozaki et al carefully characterized the selectivity of the BTK inhibitor, tirabrutinib, and its effects on phosphoproteomic in B-cell lymphoma cell lines. They also analyzed the transcriptomic changes induced by tirabrutinib in DLBCL xenograft tumors. Based on these results, the authors discussed the potential multifaceted anti-tumor mechanism of tirabrutinib. Overall, this manuscript provides interesting and important omics-data for the clinical BTK inhibitor, as well as potentially novel discoveries on the BTK signaling in DLBCL. However, I still feel that the current manuscript is not enough in-depth and need be further improved. Specifically, here are my major concerns:

1. The phosphoproteomic results should be further analyzed and sorted in terms of reported BTK substrates, reported or new direct/indirect downstream effectors, and potentially off-targets of tirabrutinib. For some important reported substrates or downstream effectors of BTK, the authors need to double-check with western blots if the proteomic studies failed to detect significant downregulation as expected, and then discuss it.

2. For the downregulated transcriptions, the key factors also need be double-checked with qPCR and western blots. Besides, the authors should provide more proof that the downregulation of IRF4 and MYC were mainly induced by suppression of NF-κB in TMD8 cells. Moreover, it would be better if the authors could analyze the transcriptomic results together with the phosphoproteomic results, and check if there is consistence for factors such as ATF2 and ERF2.

3. In Figure S1, the levels of native p-BTK were so weak that I doubt tirabrutinib inhibited p-BTK. The authors should check the phosphorylation of the direct substrate of BTK, PLCγ, and check if tirabrutinib could inhibit p-BTK after the H2O2 stimulation.

4. Tirabrutinib was much less potent to inhibit the growth of U-2932 cells, and downregulated much less phosphorylation of the proteome. The authors should further dig these findings, and check if U-2932 cells were really dependent on BTK as TMD8 cells, and look for the potentially difference of BTK signaling in both cells.

And here are several minor concerns:

1. The abstract should be written more concisely, and the English in the main text should be further improved.

2. The culturing density and the type of well plates should be added into the method for the cell viability assays.

3. Despite that tirabrutinib showed overall consistent IC50s regarding inhibition of cell growth and phospho-BTK, more evidence is needed to support the conclusion that “proliferation appears to be inhibited by tirabrutinib via inhibition of BTK phosphorylation” in line 393. Besides, tirabrutinib could not suppressed the growth of U-2932 cells even at high concentrations.

4. In line 478, the description “although the significances of all the regulated sites by tirabrutinib are unknown” is a bit confusing. Did the authors mean the functions of these phosphorylations?

Reviewer #2: In the manuscript of "nvestigation of the anti-tumor mechanism of tirabrutinib, a highly selective Bruton’s

tyrosine kinase inhibitor, by phosphoproteomics and transcriptomics" , the study is well defined. The flow of work is good starting from showing the selective of this drug on BTK among other kinases to in vivo test the efficacy of this drug.

It would be good if the author can show the structure of the complex between three generation of drugs with BTK and explain the reason why it selective against BTK than other kinases.

6. PLOS authors have the option to publish the peer review history of their article (what does this mean?). If published, this will include your full peer review and any attached files.

Reviewer #1: No

Reviewer #2: **Yes: **kiattawee choowongkomon

---

## [Author Response · Author response to Decision Letter 0]

19 Jan 2023

Journal Requirements

Response: We have ensured that our manuscript meets PLOS ONE's style requirements, including those for file naming.

"I have read the journal's policy and the authors of this manuscript have the following competing interests: This study was funded by Ono Pharmaceutical Co., Ltd. (Osaka, Japan). All authors are employees of Ono Pharmaceutical Co., Ltd." 

We note that you received funding from a commercial source: "Ono Pharmaceutical Co., Ltd."

Response: We have provided an amended Competing Interests Statement and an amended Financial Disclosure in the cover letter.

Response: There are no ethical nor legal restrictions on sharing data. We have included the minimal anonymized dataset in Supporting Information as listed below:

S2 Table. Complete KINOMEscan dataset.

S3 Table. Processed version of the MaxQuant phosphosite table of the experiments with tirabrutinib-treated TMD8 cells.

S4 Table. Processed version of the MaxQuant phosphosite table of the experiments with tirabrutinib-treated U-2932 cells.

In addition, the microarray data are available from the Gene Expression Omnibus (GEO; GSE 210284). The other datasets used and/or analyzed during the current study are available from the corresponding author on reasonable request.

Response: Original images of western blots have been provided as Supporting Information.

Responses to the comments of Reviewer #1

In this work, Kozaki et al carefully characterized the selectivity of the BTK inhibitor, tirabrutinib, and its effects on phosphoproteomic in B-cell lymphoma cell lines. They also analyzed the transcriptomic changes induced by tirabrutinib in DLBCL xenograft tumors. Based on these results, the authors discussed the potential multifaceted anti-tumor mechanism of tirabrutinib. Overall, this manuscript provides interesting and important omics-data for the clinical BTK inhibitor, as well as potentially novel discoveries on the BTK signaling in DLBCL. However, I still feel that the current manuscript is not enough in-depth and need be further improved. Specifically, here are my major concerns:

1. The phosphoproteomic results should be further analyzed and sorted in terms of reported BTK substrates, reported or new direct/indirect downstream effectors, and potentially off-targets of tirabrutinib. For some important reported substrates or downstream effectors of BTK, the authors need to double-check with western blots if the proteomic studies failed to detect significant downregulation as expected, and then discuss it.

Response: We have sorted the phosphoproteomic analysis data regarding the reported BTK substrate PLCγ2 and reported or novel direct/indirect downstream effectors in S5 Table. However, we did not conduct additional analysis and sort any data for possible off-target candidates of tirabrutinib. The reason for this is as follows:

Based on the selectivity data using in vitro cell-free kinase assays, tirabrutinib at 1 μM inhibited kinase activity of TEC, BMX, HUNK, RIPK2, ERBB4, and CSK. Among these kinases, we have demonstrated that tirabrutinib did not bind to RIPK2 and CSK in a cell-based assay using PBMC (Kd; >3 μM), which was carried out using KinAffinity (Evotec AG, Munich, Germany). These results suggest that tirabrutinib would not inhibit RIPK2 and CSK in TMD8 and U-2932 cells. As for the other kinases (TEC, BMX, HUNK, and ERBB4), there are currently no publications that report their function or contribution to cellular signaling in DLBCL cells; thus, it is difficult to discuss the contribution of the potential inhibition of these kinases by tirabrutinib with the obtained phosphoproteomic data.

Next, we have investigated the effect of tirabrutinib on some important effectors, including the BTK substrate p-PLCγ2 and p-ERK, as well as p-BTK, by western blot analysis. The data were added as S2 Fig. In both TMD8 and U-2932 cells, tirabrutinib inhibited phosphorylation at Tyr-759, which is the site reported to be phosphorylated by BTK and correlates with the lipase activity of PLCγ2 [Mol Cell Biol. 2004;24:9986-99] at a similar concentration range to that required for the suppression of BTK autophosphorylation. This indicates that tirabrutinib inhibited the phosphorylation of PLCγ2 at Y759 through BTK inhibition, although significant suppression was not detected in the phosphosites of PLCγ2 in the phosphoproteomic analysis. Regarding p-ERK, we confirmed through western blot analysis that tirabrutinib inhibited p-ERK in TMD8 cells in a concentration-dependent manner after 1 h of treatment. In U-2932 cells, treatment with tirabrutinib for 4 h resulted in significant inhibition of p-ERK in a concentration-dependent manner, although the inhibitory activity of tirabrutinib on p-ERK was not clear during 1 h of treatment. Based on those results, we confirmed the inhibitory activity of tirabrutinib on p-ERK in both phosphoproteomic and western blot analyses.

As for another downstream signaling pathway, we also checked p-PKCβ and p-p65 in NF-κB signaling in both cells, and the data were added as S3 Fig in addition to the results for p-AKT, which were already submitted as supplemental data. Regarding PKCβ, a downstream effector of PLCγ2 and upstream effector of ERK and NF-κB signaling [Mol Cancer. 2018;17:57, Front Oncol. 2020;10:591577], in the phosphoproteomic analysis, tirabrutinib did not significantly inhibit the phosphorylation at Thr-641, which is the site that is reported to be autophosphorylated and that maintains catalytic competence of PKCβ. Therefore, we tried to detect p-PKCβ in the western blot analysis but failed in both cell lines because the tested antibody did not work well. Thus, it is difficult to conclude whether tirabrutinib has an effect on the phosphorylation of PKCβ. As for NF-κB signaling, several phosphorylated sites in the related effectors, such as IKKβ and p65, were detected in the phosphoproteomic analysis. However, it was difficult to determine whether tirabrutinib suppressed NF-κB signaling because the significance of those detected sites was unclear. In the western blot analysis, however, we successfully detected p-IκBα at S32/36, which is critical for the activation of NF-κB signaling [Biochem J. 2021;478:2619-2664.]. Because tirabrutinib inhibited p-IκBα in both cell lines, we concluded that tirabrutinib showed inhibitory activity on NF-κB signaling in both cell lines.

2. For the downregulated transcriptions, the key factors also need be double-checked with qPCR and western blots. Besides, the authors should provide more proof that the downregulation of IRF4 and MYC w mainly induced by suppression of NF-κB in TMD8 cells. Moreover, it would be better if the authors could analyze the transcriptomic results together with the phosphoproteomic results, and check if there is consistence for factors such as ATF2 and ERF2.

Response: Because tirabrutinib downregulated IRF-4 and MYC pathway-related genes in the transcriptome analysis, we selected IRF4, BCL6, and MYC as key factors and evaluated the effect of tirabrutinib for expression levels of these genes and proteins after in vitro treatment with DMSO or tirabrutinib at 1 μM for 24 h in TMD8 cells by RT-PCR and western blot analysis. As a result, treatment with tirabrutinib resulted in downregulation of IRF4 and MYC genes, and upregulation of BCL6 genes, which was consistent with the microarray data. In the western blot analysis, we confirmed the downregulation of IRF4 and MYC protein expression by tirabrutinib, while a significant change in BCL6 protein was not induced by tirabrutinib. In conclusion, the results of IRF-4 and MYC pathway downregulation by tirabrutinib in the microarray experiment was confirmed with RT-PCR and western blot analysis. The obtained data and descriptions have been added to the Results section of the manuscript and S6 Fig.

Next, to clarify that the downregulation of IRF4 and MYC by tirabrutinib were mainly induced by suppression of NF-κB in TMD8 cells, we evaluated the effect of several drugs that selectively inhibit specific signaling pathways. As selective inhibitors for NF-κB, ERK, or AKT signaling, BMS-345541 (IKK inhibitor), refametinib (MEK inhibitor), and MK-2206 (AKT inhibitor) were used, respectively. As a result, treatment with BMS-345541 at 1 μM for 24 h resulted in significant downregulation in IRF4 and MYC genes. Treatment with MK-2206 at 1 μM did not downregulate these genes, and refametinib at 0.1 μM tended to downregulate only MYC genes. These results suggest that the effect of tirabrutinib on IRF4 would be mainly induced by suppression of NF-κB signaling, and the effect of tirabrutinib on MYC would be induced by suppression of not only NF-κB signaling but also ERK signaling. The obtained data and descriptions have been added to the Results section of the manuscript and S7 Fig.

Regarding your suggestion for analyses of the transcriptomic results together with the phosphoproteomic results, some descriptions were added to the Discussion section as follows:

“We also observed some correlations between the results of the phosphoproteomic and transcriptomic analyses of signaling pathways regulated by tirabrutinib in TMD8. For example, tirabrutinib significantly decreased the phosphorylation of ATF2 (Thr-71), which is phosphorylated by ERK, leading to transcriptional activation. ATF2 is reported to control ATF3 expression, which is an important element controlling the growth of ABC-DLBCL [66]. In the transcriptomic analysis, the expression of ATF3 was significantly suppressed by tirabrutinib (fold change, 0.36; P < 0.001). In addition, the observed downregulation at Thr-246 of PRAS40 by tirabrutinib in the phosphoproteomic analysis suggests the suppression of mTORC1 signaling [79], which is consistent with the downregulation of mTORC1 pathway-related genes by tirabrutinib in the transcriptomic analysis.”

3. In Figure S1, the levels of native p-BTK were so weak that I doubt tirabrutinib inhibited p-BTK. The authors should check the phosphorylation of the direct substrate of BTK, PLCγ, and check if tirabrutinib could inhibit p-BTK after the H2O2 stimulation.

Response: As shown in S2 Fig, tirabrutinib inhibited p-BTK and p-PLCγ2 in a concentration-dependent manner in both TMD8 and U-2932 cells. The concentration at which tirabrutinib inhibits p-BTK is consistent with the data shown in S1 Fig. Additionally, tirabrutinib inhibited p-PLCγ2 (through Tyr-759, a phosphosite regulated by BTK) at a similar concentration range as for p-BTK, indicating that tirabrutinib inhibits the activation of BTK (p-BTK) in TMD8 and U-2932 cells. Furthermore, by flow cytometry, we confirmed that treatment with tirabrutinib for 6 h resulted in the inhibition of p-BTK in a concentration-dependent manner with an IC50 of 9.78 nM after H2O2 stimulation in TMD8 cells, as shown below (these data are not included in the manuscript).

Figure: Effect of tirabrutinib on Btk inhibition by continuous exposure to TMD8 cells

TMD8 cells were exposed to the vehicle or various concentrations of tirabrutinib for 6 h and then stimulated with H2O2. After stimulation, the cells were stained with BTK-pY223 antibodies, and the mean fluorescence intensity (MFI) of BTK-pY223 was measured by flow cytometry. From the MFI of each sample, the increase in MFI (ΔMFI) relative to that in the negative control sample was calculated, and the inhibition rates (%) in the tirabrutinib treatment groups relative to that in the vehicle group were displayed as the mean ± standard error for three samples per group.

4. Tirabrutinib was much less potent to inhibit the growth of U-2932 cells, and downregulated much less phosphorylation of the proteome. The authors should further dig these findings, and check if U-2932 cells were really dependent on BTK as TMD8 cells, and look for the potentially difference of BTK signaling in both cells.

Response: As shown in Fig 3A, there is no significant difference in the IC50 of tirabrutinib for growth inhibition between TMD8 and U-2932, although the maximum growth inhibitions greatly differ in these cell lines. Tirabrutinib inhibited p-ERK at a similar concentration range to that of p-BTK inhibition (S2 Fig) and IC50 for growth inhibition (Fig 3A, B), which suggests that U-2932 cell growth is partially dependent on ERK signaling through p-BTK inhibition. Our phosphoproteomic and western blot analyses indicated that ERK signaling and NF-kB signaling are dependent on BTK in both TMD8 and U-2932 cell lines, whereas AKT signaling is not dependent on BTK in U-2932 cells, resulting in less downregulation of proteome phosphorylation and lower maximum growth inhibition in U-2932 compared with TMD8.

And here are several minor concerns:

5. The abstract should be written more concisely, and the English in the main text should be further improved.

Response: The Abstract and main text have been revised by a native English speaker to address the Reviewer’s concerns. We have reduced the abstract word count to 208 words (note: the PLOS ONE Abstract limit is 300 words).

6. The culturing density and the type of well plates should be added into the method for the cell viability assays.

Response: We have added information about the culturing density and well plates used to the Methods section.

7. Despite that tirabrutinib showed overall consistent IC50s regarding inhibition of cell growth and phospho-BTK, more evidence is needed to support the conclusion that “proliferation appears to be inhibited by tirabrutinib via inhibition of BTK phosphorylation” in line 393. Besides, tirabrutinib could not suppressed the growth of U-2932 cells even at high concentrations.

Response: Please see our response to comment #4. Tirabrutinib inhibited p-ERK at a similar concentration range to that of p-BTK inhibition (S2 Fig) and IC50 for growth inhibition (Fig 3A, B) in TMD8 and U-2932 cells. These data suggest that tirabrutinib exerts an anti-proliferative effect via p-ERK inhibition by p-BTK inhibition in both cell lines. Regarding the reason for the difference between TMD8 and U-2932 in the maximum inhibition for cell growth, we speculate that AKT signaling is not dependent on BTK in U-2932.

8. In line 478, the description “although the significances of all the regulated sites by tirabrutinib are unknown” is a bit confusing. Did the authors mean the functions of these phosphorylations?

Response: Yes, we were referring to the functions of all the regulated phosphosites. We have revised the description in the Results section to avoid confusion.

Responses to the comments of Reviewer #2

Reviewer #2: In the manuscript of "nvestigation of the anti-tumor mechanism of tirabrutinib, a highly selective Bruton’s tyrosine kinase inhibitor, by phosphoproteomics and transcriptomics" , the study is well defined. The flow of work is good starting from showing the selective of this drug on BTK among other kinases to in vivo test the efficacy of this drug.

1. It would be good if the author can show the structure of the complex between three generation of drugs with BTK and explain the reason why it selective against BTK than other kinases.

Response: Based on the Reviewer’s suggestion, we have added the following description in the Discussion section: 

“The binding of zanubrutinib to BTK is different from that of ibrutinib to BTK, which contributes to its superior selectivity [14]. The binding mode of tirabrutinib and acalabrutinib with BTK is similar to that of ibrutinib, but the structures of the inhibitory component are different. Butylamide electrophiles are less reactive than acrylamide, and the higher nucleophilicity of Cys-481 in BTK may explain the superior selectivity over ibrutinib [13].”

Additional revisions

In addition to the revisions outlined above, we have also made some minor amends of the text to correct some inconsistences in terminology and typographical marks. These changes are indicated using Microsoft Word’s Track Changes.

---

## [Decision Letter · Decision Letter 1]

9 Feb 2023

Investigation of the anti-tumor mechanism of tirabrutinib, a highly selective Bruton’s tyrosine kinase inhibitor, by phosphoproteomics and transcriptomics

PONE-D-22-16384R1

Dear Dr. Kozaki,

We’re pleased to inform you that your manuscript has been judged scientifically suitable for publication and will be formally accepted for publication once it meets all outstanding technical requirements.

Kind regards,

Mauro Salvi, Ph.D.

Academic Editor

PLOS ONE

Additional Editor Comments (optional):

Reviewers' comments:

Reviewer's Responses to Questions

**Comments to the Author**

1. If the authors have adequately addressed your comments raised in a previous round of review and you feel that this manuscript is now acceptable for publication, you may indicate that here to bypass the “Comments to the Author” section, enter your conflict of interest statement in the “Confidential to Editor” section, and submit your "Accept" recommendation.

Reviewer #1: All comments have been addressed

2. Is the manuscript technically sound, and do the data support the conclusions?

Reviewer #1: Yes

3. Has the statistical analysis been performed appropriately and rigorously? 

Reviewer #1: Yes

4. Have the authors made all data underlying the findings in their manuscript fully available?

Reviewer #1: Yes

5. Is the manuscript presented in an intelligible fashion and written in standard English?

Reviewer #1: Yes

6. Review Comments to the Author

Reviewer #1: (No Response)

7. PLOS authors have the option to publish the peer review history of their article (what does this mean?). If published, this will include your full peer review and any attached files.

Reviewer #1: No

---

## [Editor Report · Acceptance letter]

3 Mar 2023

PONE-D-22-16384R1 

Investigation of the anti-tumor mechanism of tirabrutinib, a highly selective Bruton’s tyrosine kinase inhibitor, by phosphoproteomics and transcriptomics 

Dear Dr. Kozaki:

I'm pleased to inform you that your manuscript has been deemed suitable for publication in PLOS ONE. Congratulations! Your manuscript is now with our production department. 

Kind regards, 

on behalf of

Prof. Mauro Salvi 

Academic Editor

PLOS ONE